# NEURAL STOCHASTIC DUAL DYNAMIC PROGRAMMING

**Hanjun Dai**[†,∗] **Yuan Xue**[◇,∗] **Zia Syed**[◇]**, Dale Schuurmans**[†]**, Bo Dai**[†]
[†]Google Research, Brain Team [◇] Google Cloud AI
{hadai, yuanxue, zsyed, schuurmans, bodai}@google.com

## ABSTRACT

Stochastic dual dynamic programming (SDDP) is a state-of-the-art method for solving multi-stage stochastic optimization, widely used for modeling real-world process optimization tasks. Unfortunately, SDDP has a worst-case complexity that scales exponentially in the number of decision variables, which severely limits applicability to only low dimensional problems. To overcome this limitation, we extend SDDP by introducing a trainable neural model that learns to map problem instances to a *piece-wise linear* value function within intrinsic *low-dimension space*, which is architected specifically to interact with a base SDDP solver, so that can accelerate optimization performance on new instances. The proposed *Neural Stochastic Dual Dynamic Programming (ν-SDDP)* continually self-improves by solving successive problems. An empirical investigation demonstrates that ν-SDDP can significantly reduce problem solving cost without sacrificing solution quality over competitors such as SDDP and reinforcement learning algorithms, across a range of synthetic and real-world process optimization problems.

## 1 INTRODUCTION

Multi-stage stochastic optimization (MSSO) considers the problem of optimizing a sequence of decisions over a finite number of stages in the presence of stochastic observations, minimizing an expected cost while ensuring stage-wise action constraints are satisfied (Birge & Louveaux, 2011; Shapiro et al., 2014). Such a problem formulation captures a diversity of real-world process optimization problems, such as asset allocation (Dantzig & Infanger, 1993), inventory control (Shapiro et al., 2014; Nambiar et al., 2021), energy planning (Pereira & Pinto, 1991), and bio-chemical process control (Bao et al., 2019), to name a few. Despite the importance and ubiquity of the problem, it has proved challenging to develop algorithms that can cope with high-dimensional action spaces and long-horizon problems (Shapiro & Nemirovski, 2005; Shapiro, 2006).

There have been a number of attempts to design scalable algorithms for MSSO, which generally attempt to exploit scenarios-wise or stage-wise decompositions. An example of a scenario-wise approach is Rockafellar & Wets (1991), which proposed a progressive hedging algorithm that decomposes the sample averaged approximation of the problem into individual scenarios and applies an augmented Lagrangian method to achieve consistency in a final solution. Unfortunately, the number of subproblems and variables grows *exponentially* in the number of stages, known as the *"curse-of-horizon"*. A similar proposal in Lan & Zhou (2020) considers a dynamic stochastic approximation, a variant of stochastic gradient descent, but the computational cost also grows exponentially in the number of stages. Alternatively, stochastic dual dynamic programming (SDDP) (Birge, 1985; Pereira & Pinto, 1991), considers a stage-wise decomposition that breaks the curse of horizon (Füllner & Rebennack, 2021) and leads to an algorithm that is often considered state-of-the-art. The method essentially applies an approximate cutting plane method that successively builds a piecewise linear convex lower bound on the optimal cost-to-go function. Unfortunately, SDDP can require an exponential number of iterations with respect to the number of decision variables (Lan, 2020), known as the *"curse-of-dimension"* (Balázs et al., 2015).

Beyond the scaling challenges, current approaches share a common shortcoming that they treat each optimization problem *independently*. It is actually quite common to solve a family of problems that share structure, which intuitively should allow the overall computational cost to be reduced (Khalil et al., 2017; Chen et al., 2019). However, current methods, after solving each problem instance via

---
[∗]Equal contribution

intensive computation, discard all intermediate results, and tackle all new problems from scratch. Such methods are destined to behave as a perpetual novice that never shows any improvement with problem solving experience.

In this paper, we present a meta-learning approach, *Neural Stochastic Dual Dynamic Programming ($\nu$-SDDP)*, that, with problem solving experience, learns to significantly improve the efficiency of SDDP in high-dimensional and long-horizon problems. In particular, $\nu$-SDDP exploits a specially designed neural network architecture that produces outputs interacting directly and conveniently with a base SDDP solver. The idea is to learn an operator that take information about the specific problem instance, and map it to a *piece-wise linear function* in the intrinsic *low-dimension* space for accurate value function approximation, so that can be plugged into a SDDP solver. The mapping is trainable, leading to an overall algorithm that self-improves as it solves more problem instances. There are three primary benefits of the proposed approach with carefully designed components:

  **i)** By adaptively generating a *low-dimension* projection for each problem instance, $\nu$-SDDP reduces the curse-of-dimension effect for SDDP.
 **ii)** By producing a reasonable value function initialization given a description of the problem instance, $\nu$-SDDP is able to *amortize* its solution costs, and gain a significant advantage over the initialization in standard SDDP on the two benchmarks studied in the paper.
**iii)** By restricting value function approximations to a *piece-wise affine form*, $\nu$-SDDP can be seamlessly incorporated into a base SDDP solver for further refining solution, which allows solution time to be reduced.

Figure 1 provides an illustration of the overall $\nu$-SDDP method developed in this paper.

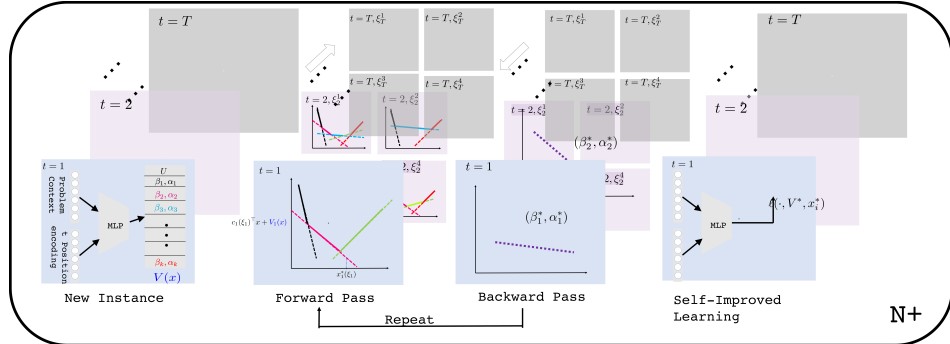

Figure 1: Overall illustration of $\nu$-SDDP. For training, the algorithm iterates N times to solve different problem instances. For each instance, it repeats two passes: forward (solving LPs to estimate an optimal action sequence) and backward (adding new affine components to the value function estimate). Once a problem instance is solved, the optimal value function and optimal actions are used for neural network training. During inference time for a new problem, it can predict high-quality value function with little cost, which can be embedded into SDDP for further improvements.

The remainder of the paper is organized as follows. First, we provide the necessary background on MSSO and SDDP in Section 2. Motivated by the difficulty of SDDP and shortcomings of existing learning-based approaches, we then propose $\nu$-SDDP in Section 3, with the design of the neural component and the learning algorithm described in Section 3.1 and Section 3.2 respectively. We compare the proposed approach with existing algorithms that also exploit supervised learning (SL) and reinforcement learning (RL) for MSSO problems in Section 4. Finally, in Section 5 we conduct an empirical comparison on synthetic and real-world problems and find that $\nu$-SDDP is able to effectively exploit successful problem solving experiences to greatly accelerate the planning process while maintaining the quality of the solutions found.

## 2 PRELIMINARIES

We begin by formalizing the multi-stage stochastic optimization problem (MSSO), and introducing the stochastic dual dynamic programming (SDDP) strategy we exploit in the subsequent algorithmic development. We emphasize the connection and differences between MSSO and a Markov decision process (MDP), which shows the difficulties in applying the advanced RL methods for MSSO.

**Multi-Stage Stochastic Optimization (MSSO).** Consider a multi-stage decision making problem with stages $t = 1, \ldots, T$, where an *observation* $\xi_t \sim P_t(\cdot)$ is drawn at each stage from a known

observation distribution $P_t$. A full observation history $\{\xi_t\}_{t=1}^T$ forms a *scenario*, where the observations are assumed independent between stages. At stage $t$, an *action* is specified by a vector $x_t$. The goal is to choose a sequence of actions $\{x_t\}_{t=1}^T$ to minimize the overall expected sum of linear costs $\sum_{t=1}^T c_t(\xi_t)^\top x_t$ under a known *cost function* $c_t$. This is particularly challenging with feasible constraints on the action set. Particularly, the *feasible action set* $\chi_t$ at stage $t$ is given by

$$\chi_t(x_{t-1}, \xi_t) \ := \ \{x_t | A_t(\xi_t) x_t = b_t(\xi_t) - B_{t-1}(\xi_t) x_{t-1}, x_t \geqslant 0\}, \quad \forall t = 2, \ldots, T, \quad (1)$$

$$\chi_1(\xi_1) \ := \ \{x_1 | A_1(\xi_1) x_1 = b_1(\xi_1), x_1 \geqslant 0\}, \quad (2)$$

where $A_t$, $B_t$ and $b_t$ are known functions. Notably, the feasible set $\chi_t(x_{t-1}, \xi_t)$ at stage $t$ depends on the previous action $x_{t-1}$ and the current stochastic observation $\xi_t$. The MSSO problem can then be expressed as

$$v := \begin{cases} \min_{x_1} c_1(\xi_1)^\top x_1 + \mathbb{E}_{\xi_2} \left[ \min_{x_2} c_2(\xi_2)^\top x_2 \cdots + \mathbb{E}_{\xi_T} \left[ \min_{x_T} c_T(\xi_T)^\top x_T \right] \right], \\ \text{s.t.} \quad x_t \in \chi_t(x_{t-1}, \xi_t) \quad \forall t = \{1, \ldots, T\}, \end{cases} \quad (3)$$

where $\xi_1$ and the *problem context* $U = \{u_t\}_{t=1}^T$ for $u_t := (P_t, c_t, A_t, B_t, b_t)$ are provided. Given the context $U$ given, the MSSO is specified. Since the elements in $U$ are probability or functions, the definition of $U$ is only conceptual. In practice, we implement $U$ with its sufficient representations. We will demonstrate the instantiation of $U$ in our experiment section. MSSO is often used to formulate real-world inventory control and portfolio management problems; we provide formulations for these specific problems in in Appendix B.1 and Appendix B.2.

Similar to MDPs, *value functions* provide a useful concept for capturing the structure of the optimal solution in terms of a temporal recurrence. Following the convention in the MSSO literature (Füllner & Rebennack, 2021), let

$$Q_t(x_{t-1}, \xi_t) := \min_{x_t \in \chi_t(x_{t-1}, \xi_t)} c_t(\xi_t)^\top x_t + \underbrace{\mathbb{E}_{\xi_{t+1}}[Q_{t+1}(x_t, \xi_{t+1})]}_{V_{t+1}(x_t)}, \quad \forall t = 2, \ldots, T, \quad (4)$$

which expresses a Bellman optimality condition over the feasible action set. Using this definition, the MSSO problem (3) can then be rewritten as

$$v := \{\min_{x_1} c_1(\xi_1)^\top x_1 + V_2(x_1), \quad \text{s.t.} \ x_1 \in \chi_1(\xi_1)\}. \quad (5)$$

**Theorem 1 (informal, Theorem 1.1 and Corollary 1.2 in Füllner & Rebennack (2021))** *When the optimization (5) almost surely has a feasible solution for every realized scenario, the value functions $Q_t(\cdot, \xi)$ and $V_t(\cdot)$ are piecewise linear and convex in $x_t$ for all $t = 1, \ldots, T$.*

**Stochastic Dual Dynamic Programming (SDDP).** Given the problem specification (3) we now consider solution strategies. *Stochastic dual dynamic programming* (SDDP) (Shapiro et al., 2014; Füllner & Rebennack, 2021) is a state-of-the-art approach that exploits the key observation in Theorem 1 that the optimal $V$ function can be expressed as a maximum over a finite number of linear components. Given this insight, SDDP applies Bender's decomposition

---

**Algorithm 1** $\text{SDDP}(\{V_t^0\}_{t=1}^T, \xi_1, n)$

1: Sample $\{\xi_t^j\}_{j=1}^m \sim P_t(\cdot)$ for $t = 2, \ldots, T$
2: **for** $i = 1, \ldots, n$ **do**
3:      Select $J$ samples from *uniform*$\{1, ..., m\}$    ▷ minibatch
4:      **for** $t = 1, \ldots, T$ and $j = 1, \ldots, J$ **do**    ▷ forward pass
5:
$$x_{tj}^i \in \begin{cases} \arg\min c_t(\xi_t^j)^\top x_t + V_{t+1}^i(x_t), \\ \text{s.t. } x_{tj} \in \chi_t(x_{t-1,j}^i, \xi_t^j) \end{cases} \quad (6)$$
6:      **end for**
7:      **for** $t = T, \ldots, 1$ **do**    ▷ backward pass
8:        Calculate the dual variables of (6) for each $\xi_t^j$ in (29);
9:        Update $V_t^i$ with dual variables via (32) (Appendix C)
10:      **end for**
11: **end for**

---

to the sample averaged approximation of (3). In particular, it performs two steps in each iteration: **(i)** in a *forward pass* from $t = 0$, trial solutions for each stage are generated by solving subproblems (4) using the current estimate of the future expected-cost-to-go function $V_{t+1}$; **(ii)** in a *backward pass* from $t = T$, each of the $V$-functions are then updated by adding cutting planes derived from the optimal actions $x_{t-1}$ obtained in the forward pass. (Details for the cutting plan derivation and the connection to TD-learning are given in Appendix C due to lack of space.) After each iteration, the current $V_{t+1}$ provides a lower bound on the true optimal expected-cost-to-go function, which is being successively tightened; see Algorithm 1.

**MSSO vs. MDPs.** At the first glance, the dynamics in MSSO (3) is describing markovian relationship on *actions*, *i.e.*, the current action $x_t$ is determined by current state $\xi_t$ and previous action $x_{t-1}$, which is different from the MDPs on markovian *states*. However, we can equivalently reformulate MSSO as

a MDP with state-dependent feasible action set, by defining the $t$-th step state as $s_t := (x_{t-1}, \xi_t)$, and action as $a_t := x_t \in \chi_t(x_{t-1}, \xi_t)$. This indeed leads to the markovian transition $p_{\text{set}}(s_{t+1}|s_t, x_t) = \mathbf{1}(x_t \in \chi_t(x_{t-1}, \xi_t)) p_{t+1}(\xi_{t+1})$, where $\mathbf{1}(x_t \in \chi_t(x_{t-1}, \xi_t)) := \{1 \text{ if } x_t \in \chi_t(x_{t-1}, \xi_t), 0 \text{ otherwise}\}$.

Although we can represent MSSO equivalently in MDP, the MDP formulation introduces extra difficulty in maintaining state-dependent feasible action and ignores the linear structure in feasibility, which may lead to the inefficiency and infeasibility when applying RL algorithms (see Section 5). Instead MSSO take these into consideration, especially the feasibility.

Unfortunately, MSSO and MDP comes from different communities, the notational conventions of the MSSO versus MDP literature are *directly contradictory*: the $Q$-function in (4) corresponds to the state-value $V$-function in the MDP literature, whereas the $V$-function in (4) is particular to the MSSO setting, integrating out of randomness in state-value function, which has no standard correspondent in the MDP literature. In this paper, *we will adopt the notational convention of MSSO*.

## 3    Neural Stochastic Dual Dynamic Programming

Although SDDP is a state-of-the-art approach that is widely deployed in practice, it does not scale well in the dimensionality of the action space (Lan, 2020). That is, as the number of decision variables in $x_t$ increases, the number of generated cutting planes in the $V_{t+1}$ approximations tends to grow exponentially, which severely limits the size of problem instance that can be practically solved. To overcome this limitation, we develop a new approach to scaling up SDDP by leveraging the generalization ability of deep neural networks across different MSSO instances in this section.

We first formalize the learning task by introducing the **contextual MSSO**. Specifically, as discussed in Section 2, the problem context $U = \{u_t\}_{t=1}^T$ with $u_t := (P_t, c_t, A_t, B_t, b_t)$ soly defines the MSSO problem, therefore, we denote $\mathcal{W}(U)$ as an instance of MSSO (3) with explicit dependence on $U$. We assume the MSSO samples can be instantiated from contextual MSSO following some distribution, *i.e.*, $\mathcal{W}(U) \sim \mathcal{P}(\mathcal{W})$, or equivalently, $U \sim \mathcal{P}(U)$. Then, instead of treating each MSSO *independently* from scratch, we can learn to amortize and generalize the optimization across different MSSOs in $\mathcal{P}(\mathcal{W})$. We develop a meta-learning strategy where a model is trained to map the $u_t$ and $t$ to a piecewise linear convex $V_t$-approximator that can be directly used to initialize the SDDP solver in Algorithm 1. In principle, if optimal value information can be successfully transferred between similar problem contexts, then the immense computation expended to recover the optimal $V_t$ functions for previous problem contexts can be leveraged to shortcut the nearly identical computation of the optimal $V_t$ functions for a novel but similar problem context. In fact, as we will demonstrate below, such a transfer strategy proves to be remarkably effective.

### 3.1    Neural Architecture for Mapping to Value Function Representations

To begin the specific development, we consider the structure of the value functions, which are desired in the deep neural approximator.

- **Small approximation error and easy optimization over action:** recall from Theorem 1 that the optimal $V_t$-function must be a convex, piecewise linear function. Therefore, it is sufficient for the output representation from the deep neural model to express max-affine function approximations for $V_t$, which conveniently are also directly usable in the minimization (6) of the SDDP solver.
- **Encode the instance-dependent information:** to ensure the learned neural mapping can account for instance specific structure when transferring between tasks, the output representation needs to encode the problem context information, $\{(P_t, c_t, A_t, B_t, b_t)\}_{t=1}^T$.
- **Low-dimension representation of state and action:** the complexity of subproblems in SDDP depends on the dimension of the state and action exponentially, therefore, the output $V_t$-function approximations should only depend on a low-dimension representation of $x$.

For the first two requirements, we consider a deep neural representation for functions $f(\cdot, u_t) \in \mathcal{M}^K$ for $t = 1, \ldots, T$, where $\mathcal{M}^K$ is the piece-wise function class with $K$ components, *i.e.*,

$$\mathcal{M}^K := \left\{ \phi(\cdot) : \mathcal{X} \to \mathbb{R} \big| \phi(x) = \max_{k=1,\ldots,K} \beta_k^\top x + \alpha_k, \beta_k \in \mathbb{R}^d, \alpha_k \in \mathbb{R} \right\}. \quad (7)$$

That is, such a function $f$ takes the problem context information $u$ as input, and outputs a set of parameters $(\{\alpha_k\}, \{\beta_k\})$ that define a max-affine function $\phi$. We emphasize that although we consider $\mathcal{M}^K$ with fixed number of linear components, it is straightforward to generalize to the function class with *context-dependent* number of components via introducing learnable $K(u_t) \in \mathbb{N}$. A key property of this output representation is that it always remains within the set of valid $V$-functions, therefore, it can be naturally incorporated into SDDP as a warm start to refine the solution. This approach leaves design flexibility around the featurization of the problem context $U$ and actions $x$, while enabling the use of neural networks for $f(\cdot, u)$, which can be trained end-to-end.

To further achieve a low-dimensional dependence on the action space while retaining convexity of the value function representations for the third desideratum, we incorporate a linear projection $x = Gy$ with $y \in \mathbb{R}^p$ and $p < d$, such that $G = \psi(u)$ satisfies $G^\top G = \mathbf{I}$. With this constraint, the mapping $f(\cdot, u)$ will be in:

$$\mathcal{M}_G^K := \left\{ \phi(\cdot) : \mathcal{Y} \to \mathbb{R} \middle| \phi_G(y) = \max_{k=1,\ldots,K} \beta_k^\top Gy + \alpha_k, \beta_k \in \mathbb{R}^d, G \in \mathbb{R}^{d \times p}, \alpha_k \in \mathbb{R} \right\}. \quad (8)$$

We postpone the learning of $f$ and $\psi$ to Section 3.2 and first illustrate the accelerated SDDP solver in the learned effective dimension of the action space in Algorithm 2. Note that after we obtain the solution $y$ in line 3 in Algorithm 2 of the projected problem, we can recover $x = Gy$ as a coarse solution for fast inference. If one wanted a more refined solution, the full SDDP could be run on the un-projected instance starting from the updated algorithm state after the fast call.

**Practical representation details:** In our implementation, we first encode the index of time step $t$ by a positional encoding (Vaswani et al., 2017) and exploit sufficient statistics to encode the distribution $P(\xi)$ (assuming in addition that the $P_t$ are stationary). As the functions $c_t$, $A_t$, $B_t$ and $b_t$ are typically static and problem specific, there structures will remain the same for different $P_t$. In our paper we focus on

---

**Algorithm 2** Fast-Inference($\{u_t\}_{t=1}^T, f, \psi, \xi_1$)

1: Set $G = \psi(U)$         ▷ fast inference
2: Projected problem instance $\{q_t\}_{t=1}^T = \{Gu_t\}_{t=1}^T$,
3: $\left\{ \tilde{y}_t\left(\xi_t^j\right) \right\}_{t,j} = \text{SDDP}\left(\{f(\cdot, q_t)\}_{t=1}^T, \xi_1, 1\right),$   ▷ we only need **one forward pass** in low-dimension space.
4: $\left\{ \tilde{x}_t\left(\xi_t^j\right) \right\}_{t,j} = \left\{ G\tilde{y}_t\left(\xi_t^j\right) \right\}_{t,j},$
/* Optional refinement */
5: $\left\{ x_t^*(\xi_t^j) \right\}_{t,j} = \text{SDDP}\left(\{f(\cdot, u_t)\}_{t=1}^T, \xi_1, n\right)$ ▷ refine solution.

---

the generalization within a problem type (*e.g.*, the inventory management) and do not expect the generalization across problems (*e.g.*, train on portfolio management and deploy on inventory management). Hence we can safely ignore these LP specifications which are typically of high dimensions. The full set of features are vectorized, concatenated and input to a 2-layer MLP with 512-hidden relu neurons. The MLP outputs $k$ linear components, $\{\alpha_k\}$ and $\{\beta_k\}$, that form the piece-wise linear convex approximation for $V_t$. For the projection $G$, we simply share one $G$ across all tasks, although it is not hard to incorporate a neural network parametrized $G$. The overall deep neural architecture is illustrated in Figure 5 in Appendix D.

## 3.2 META SELF-IMPROVED LEARNING

The above architecture will be trained using a meta-learning strategy, where we collect successful prior experience $\mathcal{D}_n := \left\{ z_i := \left( U, \{V_t^*\}_{t=1}^T, \{x_t^*(\xi_j)\}_{t=1,j}^{T,m} \right)_i \right\}_{i=1}^n$ by using SDDP to solve a set of training problem instances. Here, $U = \{u_t\}_{t=1}^T$ denotes a problem context, and $\{V_t^*\}_{t=1}^T$ and $\{x_{tj}^*\}_{t,j}^{T,m}$ are the optimal value functions and actions obtained by SDDP for each stage.

Given the dataset $\mathcal{D}_n$, the parameters of $f$ and $\psi$, $W := \{W_f, W_\psi\}$, can be learned by optimizing the following objective via stochastic gradient descent (SGD):

$$\min_W \sum_{z \in \mathcal{D}_n} \ell(W; z) := \sum_{i=1}^n \sum_{t=1}^T \left( -\sum_j^m (x_{tj}^{i*})^\top G_\psi^i \left(G_\psi^i\right)^\top x_{tj}^{i*} + \text{EMD}\left(f(\cdot; u_t^i), V_t^{i*}(\cdot)\right) \right) + \lambda \sigma(W),$$

$$\text{s.t.} \left(G_\psi^i\right)^\top G_\psi^i = I_p, \quad \forall i = 1, \ldots, n \quad (9)$$

where $G_\psi := \psi(u)$, $\text{EMD}(f, V)$ denotes the *Earth Mover's Distance* between $f$ and $V$, and $\sigma(W)$ denotes a convex regularizer on $W$. Note that the loss function (9) is actually seeking

---

**Algorithm 3** $\nu$-SDDP

---

1: Initialize dataset $\mathcal{D}_0$;
2: **for** epoch $i = 1, \ldots, n$ **do**
3:     Sample a multi-stage stochastic decision problems $U = \{u_t\}_{t=1}^T \sim P(U)$;
4:     Initial $\left\{V_t^0\right\}_{t=1}^T = (1-\gamma)\mathbf{0} + \gamma \left\{ f_W\left(\cdot, \{u_t\}_{t=1}^T\right) \right\}_{t=0}^T$ with $\gamma \sim \mathcal{B}(p_i)$;
5:     $\left( \left\{ x^*(\xi_t^j) \right\}_{t,j=1}^{T,m} \right) = \text{SDDP}(\left\{V_t^0\right\}_{t=1}^T, \xi_1, n)$;
6:     Collect solved optimization instance $\mathcal{D}_i = \mathcal{D}_{i-1} \cup \left( U, \{V_t^*\}_{t=1}^T, \{x_t^*(\xi_j)\}_{t=1,j}^{T,m} \right)$;
7:     **for** iter $= 1, \ldots, b$ **do**
8:         Sample $z_l \sim \mathcal{D}_i$;
9:         Update parameters $W$ with stochastic gradients: $W = W - \eta \nabla_W \ell(W; z_l)$;
10:     **end for**
11: **end for**

---

to maximize $\sum_{i=1}^n \sum_{t=1}^t \sum_j^m (x_{tj}^{i*})^\top G_\psi G_\psi^\top x_{tj}^{i*}$ under orthonormality constraints, hence it seeks principle components of the action spaces to achieve dimensionality reduction.

To explain the role of EMD, recall that $f(x, u)$ outputs a convex piecewise linear function represented by $\left\{ (\beta_k^f)^\top x + \alpha_k^f \right\}_{k=1}^K$, while the optimal value function $V_t^*(x) := \left\{ (\beta_l^*)^\top x + \alpha_l^*(\xi) \right\}_{l=1}^t$ in SDDP is also a convex piecewise linear function, hence expressible by a maximum over affine functions. Therefore, EMD$(f, V_t^*)$ is used to calculate the distance between the sets $\left\{ \beta_k^f, \alpha_k^f \right\}_{k=1}^K$ and $\left\{ \beta_l^*, \alpha_l^* \right\}_{l=1}^t$, which can be recast as

$$\min_{M \in \Omega(K,t)} \langle M, D \rangle, \quad \Omega(K, t) = \left\{ M \in \mathbb{R}_+^{K \times t} | M\mathbf{1} \leqslant 1, M^\top \mathbf{1} \leqslant 1, \mathbf{1}^\top M \mathbf{1} = \min(K, t) \right\}, \quad (10)$$

where $D \in \mathbb{R}^{K \times t}$ denotes the pairwise distances between elements of the two sets. Due to space limits, please refer to Figure 6 in Appendix D for an illustration of the overall training setup. The main reason we use EMD is due to the fact that $f$ and $V^*$ are *order invariant*, and EMD provides an optimal transport comparison (Peyré et al., 2019) in terms of the minimal cost over all pairings.

**Remark (Alternative losses):** One could argue that it suffices to use the vanilla regression losses, such as the $L_2$-square loss $\|f(\cdot, u, \xi) - V^*(\cdot, \xi)\|_2^2$, to fit $f$ to $V^*$. However, there are several drawbacks with such a direct approach. First, such a loss ignores the inherent structure of the functions. Second, to calculate the loss, the observations $x$ are required, and the optimal actions from SDDP are not sufficient to achieve a robust solution. This approach would require an additional sampling strategy that is not clear how to design (Defourny et al., 2012).

**Training algorithm:** The loss (9) pushes $f$ to approximate the optimal value functions for the training contexts, while also pushing the subspace $G_\psi$ to acquire principle components in the action space. The intent is to achieve an effective approximator for the value function in a low-dimensional space that can be used to warm-start SDDP inference Algorithm 2. Ideally, this should result in an efficient optimization procedure with fewer optimization variables that can solve a problem instance with fewer forward-backward passes. In an on-line deployment, the learned components, $f$ and $\psi$, can be continually improved from the results of previous solves. One can also optionally exploit the learned component for the initialization of value function in SDDP by annealing with a mixture of zero function, where the weight is sampled from a Bernolli distribution. Overall, this leads to the Meta Self-Improved SDDP algorithm, $\nu$-SDDP, shown in Algorithm 3.

## 4 RELATED WORK

The importance of MSSO and the inherent difficulty of solving MSSO problems at a practical scale has motivated research on **hand-designed approximation** algorithms, as discussed in Appendix A. **Learning-based MSSO approximations** have attracted more attentions. Rachev & Römisch (2002); Høyland et al. (2003); Hochreiter & Pflug (2007) learn a sampler for generating a small scenario tree while preserving statistical properties. Recent advances in RL is also exploited. Defourny et al. (2012) imitate a parametrized policy that maps from scenarios to actions from some SDDP solvers. Direct policy improvement from RL have also been considered. Ban & Rudin (2019) parametrize a policy as a linear model in (25), but introducing large approximation errors. As an extension, Bertsimas & Kallus (2020); Oroojlooyjadid et al. (2020) consider more complex function approximators for the policy parameterization in (25). Oroojlooyjadid et al. (2021); Hubbs et al. (2020); Balaji et al.

(2019); Barat et al. (2019) directly apply deep RL methods. Avila et al. (2021) exploit off-policy RL tricks for accelerating the SDDP $Q$-update. More detailed discussion about Avila et al. (2021) can be found in Appendix A. Overall, the majority of methods are not able to easily balance MSSO problem structures and flexibility while maintaining strict feasibility with efficient computation. They also tend to focus on learning a policy for a single problem, which does not necessarily guarantee effective generalization to new cases, as we find in the empirical evaluation.

**Context-based meta-RL** is also relevant, where the context-dependent policy (Hausman et al., 2018; Rakelly et al., 2019; Lan et al., 2019) or context-dependent value function (Fakoor et al., 2019; Arnekvist et al., 2019; Raileanu et al., 2020) is introduced. Besides the difference in MSSO vs. MDP in Section 2, the most significant difference is the parameterization and inference usage of context-dependent component. In $\nu$-SDDP, we design the specific neural architecture with the output as a piece-wise linear function, which takes the structure of MSSO into account and can be seamlessly integrated with SDDP solvers for further solution refinement with the *feasibility guaranteed*; while in the vanilla context-based meta-RL methods, the context-dependent component with arbitrary neural architectures, which will induce extra approximation error, and is unable to handle the constraints. Meanwhile, the design of the neural component in $\nu$-SDDP also leads to our particular learning objective and stochastic algorithm, which exploits the inherent piece-wise linear structure of the functions, meanwhile bypasses the additional sampling strategy required for alternatives.

## 5 EXPERIMENTS

**Problem Definition.** We first tested on **inventory optimization** with the problem configuration in Table. 1. We break the problem contexts into two sets: 1) *topology*, parameterized via the number of suppliers $S$, inventories $I$, and customers $C$; 2) *decision horizon* $T$. Note

| Problem Setting | Configuration ($S$-$I$-$C$, $T$) |
|---|---|
| Small-size topology, Short horizon (Sml-Sht) | 2-2-4, 5 |
| Mid-size topology, Long horizon (Mid-Lng) | 10-10-20, 10 |
| Portfolio Optimization | $T = 5$ |

Table 1: Problem Configuration in Inventory Optimization.

that in the Mid-Lng setting there are 310 continuous action variables, which is of magnitudes larger than the ones used in inventory control literature (Graves & Willems, 2008) and benchmarks, *e.g.*, ORL (Balaji et al., 2019) or meta-RL (Rakelly et al., 2019) on MuJoCo (Todorov et al., 2012).

Within each problem setting, a problem instance is further captured by the problem context. In inventory optimization, a forecast model is usually used to produce continuous demand forecasts and requires re-optimization of the inventory decisions based on the new distribution of the demand forecast, forming a group of closely related problem instances. We treat the parameters of the demand forecast as the primary problem context. In the experiment, demand forecasts are synthetically generated from a normal distribution: $\mathbf{d}_t \sim \mathcal{N}(\mu_{\mathbf{d}}, \sigma_{\mathbf{d}})$. For both problem settings, the mean and the standard deviation of the demand distribution are sampled from the meta uniform distributions: $\mu_{\mathbf{d}} \sim \mathcal{U}(11, 20)$, $\sigma_{\mathbf{d}} \sim \mathcal{U}(0, 5)$. Transport costs from inventories to customers are also subject to frequent changes. We model it via a normal distribution: $\mathbf{c}_t \sim \mathcal{N}(\mu_{\mathbf{c}}, \sigma_{\mathbf{c}})$ and use the distribution mean $\mu_{\mathbf{c}} \sim \mathcal{U}(0.3, 0.7)$ as the secondary problem context parameter with fixed $\sigma_{\mathbf{c}} = 0.2$. Thus in this case, the context for each problem instance that $f(\cdot, \mu_t)$ needs to care about is $u_t = (\mu_d, \sigma_d, \mu_c)$.

The second environment is **portfolio optimization**. A forecast model is ususally used to produce updated stock price forcasts and requires re-optimization of asset allocation decisions based on the new distribution of the price forecast, forming a group of closely related problem instances. We use an autoregressive process of order 2 to learn the price forecast model based on the real daily stock prices in the past 5 years. The last two-day historical prices are used as problem context parameters in our experiments. In this case the stock prices of first two days are served as the context $U$ for $f$.

Due to the space limitation, we postpone the detailed description of problems and additional performances comparison in Appendix E.

**Baselines.** In the following experiments, we compare $\nu$-SDDP with mainstream methodologies:

- **SDDP-optimal:** This is the SDDP solver that runs on each test problem instance until convergence, and is expected to produce the best solution and serve as the ground-truth for comparison.
- **SDDP-mean:** It is trained once based on the mean values of the problem parameter distribution, and the resulting $V$-function will be applied in all different test problem instances as a surrogate of the true $V$-function. This approach enjoys the fast runtime time during inference, but would yield suboptimal results as it cannot adapt to the change of the problem contexts.

| Task | Parameter Domain | SDDP-mean | $\nu$-SDDP-fast | $\nu$-SDDP-accurate | Best RL |
|---|---|---|---|---|---|
| Sml-Sht | demand mean ($\mu_{\mathbf{d}}$) | $16.15 \pm 18.61\%$ | $2.42 \pm 1.84\%$ | $\mathbf{1.32 \pm 1.13\%}$ | $38.42 \pm 17.78\%$ |
| | joint ($\mu_{\mathbf{d}}$ & $\sigma_{\mathbf{d}}$) | $20.93 \pm 22.31\%$ | $4.77 \pm 3.80\%$ | $\mathbf{1.81 \pm 2.19\%}$ | $33.08 \pm 8.05\%$ |
| Mid-Long | demand mean ($\mu_{\mathbf{d}}$) | $24.77 \pm 27.04\%$ | $2.90 \pm 1.11\%$ | $\mathbf{1.51 \pm 1.08\%}$ | $17.81 \pm 10.26\%$ |
| | joint ($\mu_{\mathbf{d}}$ & $\sigma_{\mathbf{d}}$) | $27.02 \pm 29.04\%$ | $5.16 \pm 3.22\%$ | $\mathbf{3.32 \pm 3.06\%}$ | $50.19 \pm 5.57\%$ |
| | joint ($\mu_{\mathbf{d}}$ & $\sigma_{\mathbf{d}}$ & $\mu_{\mathbf{c}}$) | $29.99 \pm 32.33\%$ | $7.05 \pm 3.60\%$ | $\mathbf{3.29 \pm 3.23\%}$ | $135.78 \pm 17.12\%$ |

Table 2: Average Error Ratio of Objective Value.

| Task | Parameter Domain | SDDP-optimal | SDDP-mean | $\nu$-SDDP-fast | $\nu$-SDDP-accurate | Best RL |
|---|---|---|---|---|---|---|
| Sml-Sht | demand mean ($\mu_{\mathbf{d}}$) | $6.80 \pm 7.45$ | $14.83 \pm 17.90$ | $9.60 \pm 3.35$ | $10.12 \pm 4.03$ | $\mathbf{3.90 \pm 8.39}$ |
| | joint ($\mu_{\mathbf{d}}$ & $\sigma_{\mathbf{d}}$) | $10.79 \pm 19.75$ | $19.83 \pm 22.02$ | $11.04 \pm 10.83$ | $13.73 \pm 16.64$ | $\mathbf{1.183 \pm 4.251}$ |
| Mid-Long | demand mean ($\mu_{\mathbf{d}}$) | $51.96 \pm 14.90$ | $73.39 \pm 59.90$ | $44.27 \pm 9.00$ | $33.42 \pm 18.01$ | $\mathbf{1.98 \pm 2.65}$ |
| | joint ($\mu_{\mathbf{d}}$ & $\sigma_{\mathbf{d}}$) | $54.89 \pm 32.35$ | $85.76 \pm 77.62$ | $45.53 \pm 24.14$ | $\mathbf{36.31 \pm 20.49}$ | $205.51 \pm 150.90$ |
| | joint ($\mu_{\mathbf{d}}$ & $\sigma_{\mathbf{d}}$ & $\mu_{\mathbf{c}}$) | $55.14 \pm 38.93$ | $86.26 \pm 81.14$ | $44.80 \pm 28.57$ | $\mathbf{36.19 \pm 20.08}$ | $563.19 \pm 114.03$ |

Table 3: Objective Value Variance.

- **Model-free RL algorithms:** Four RL algorithms, including DQN, DDPG, SAC, PPO, are directly trained online on the test instances without the budget limit of number of samples. So this setup has more privileges compared to typical meta-RL settings. We only report the best RL result in Table 2 and Table 3 due to the space limit. Detailed hyperparameter tuning along with the other performance results are reported in Appendix E.

- **$\nu$-SDDP-fast:** This is our algorithm where the the meta-trained neural-based $V$-function is directly evaluated on each problem instance, which corresponds to Algorithm 2 without the last refinement step. In this case, only one forward pass of SDDP using the neural network predicted $V$-function is needed and the $V$-function will not be updated. The only overhead compared to SDDP-mean is the feed-forward time of neural network, which can be ignored compared to the expensive LP solving.

- **$\nu$-SDDP-accurate:** It is our full algorithm presented in Algorithm 2 where the meta-trained neural-based $V$-function is further refined with 10 more iterations of vanilla SDDP algorithm.

### 5.1 SOLUTION QUALITY COMPARISON

For each new problem instance, we evaluate the algorithm performance by solving and evaluating the optimization objective value using the trained $V$-function model over 50 randomly sampled trajectories. We record the mean(**candidate**) and the standard derivation of these objective values produced by each **candidate** method outlined above. As SDDP-optimal is expected to produce the best solution, we use its mean on each problem instance to normalize the difference in solution quality. Specifically, *error ratio* of method **candidate** with respect to SDDP-optimal is:

$$\phi = \frac{\text{mean}(\textbf{candidate}) - \text{mean}(\textbf{SDDP-optimal})}{\text{abs}\{\text{mean}(\textbf{SDDP-optimal})\}} \tag{11}$$

**Inventory optimization:** We report the average optimalty ratio of each method on the held-out test problem set with 100 instances in Table 2. By comparison, $\nu$-SDDP learns to adaptive to each problem instance, and thus is able to outperform these baselines by a significantly large margin. Also we show that by tuning the SDDP with the $V$-function initialized with the neural network generated cutting planes for just 10 more steps, we can further boost the performance ($\nu$-SDDP-accurate). In addition, despite the recent reported promising results in applying deep RL algorithms in small-scale inventory optimization problems (Bertsimas & Kallus, 2020; Oroojlooyjadid et al., 2020; 2021; Hubbs et al., 2020; Balaji et al., 2019; Barat et al., 2019), it seems that these algorithms get worse results than SDDP and $\nu$-SDDP variants when the problem size increases.

We further report the average variance along with its standard deviation of different methods in Table 3. We find that generally our proposed $\nu$-SDDP (both fast and accurate variants) can yield solutions with comparable variance compared to SDDP-optimal. SDDP-mean gets higher variance, as its performance purely depends on how close the sampled problem parameters are to their means.

**Portfolio optimization:** We evaluated the same metrics as above. We train a multi-dimensional second-order autoregressive model for the selected US stocks over last 5 years as the price forecast model, and use either synthetic (low) or estimated (high) variance of the price to test different models. When the variance is high, the best policy found by SDDP-optimal is to buy (with appropriate but different asset allocations at different days) and hold for each problem instance. We found our

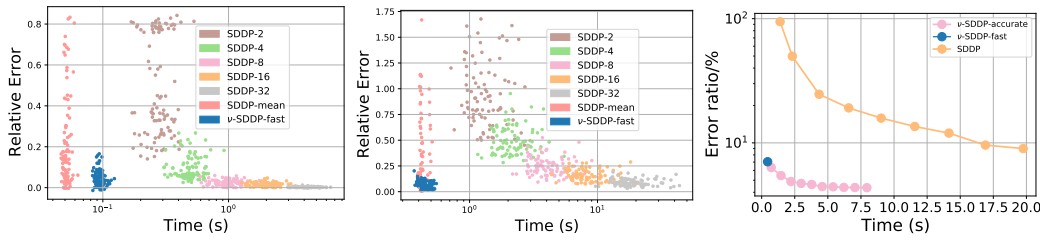

Sml-Sht-joint ($\mu_{\mathbf{d}}$ & $\sigma_{\mathbf{d}}$)  Mid-Lng-joint ($\mu_{\mathbf{d}}$ & $\sigma_{\mathbf{d}}$ & $\mu_{\mathbf{c}}$) Mid-Lng-joint ($\mu_{\mathbf{d}}$ & $\sigma_{\mathbf{d}}$ & $\mu_{\mathbf{c}}$)

Figure 2: Time-solution trade-off. In the left two plots, each dot represents a problem instance with the runtime and the solution quality obtained by corresponding algorithm. The right most plot shows how $\nu$-SDDP-accurate improves further when integrated into SDDP solver.

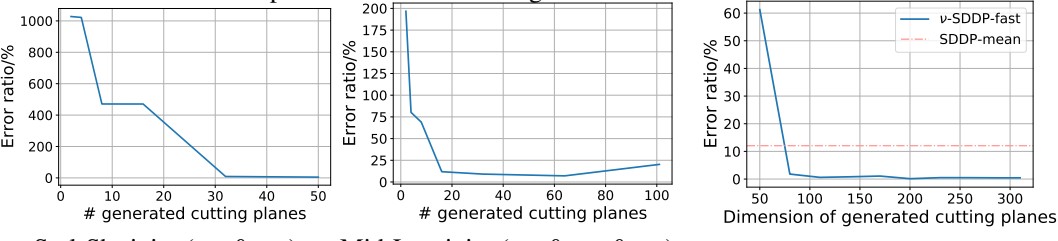

Sml-Sht-joint ($\mu_{\mathbf{d}}$ & $\sigma_{\mathbf{d}}$) Mid-Lng-joint ($\mu_{\mathbf{d}}$ & $\sigma_{\mathbf{d}}$ & $\mu_{\mathbf{c}}$)

Figure 4: Performance of $\nu$-SDDP with low-rank projection.

Figure 3: $\nu$-SDDP-fast with different # generated cutting planes. $\nu$-SDDP is able to rediscover this policy; when the variance is low, our model is also able to achieve much lower error ratio than SDDP-mean. We provide study details in Appendix E.2.

## 5.2 TRADE-OFF BETWEEN RUNNING TIME AND ALGORITHM PERFORMANCE

We study the trade-off between the runtime and the obtained solution quality in Figure 2 based on the problem instances in the test problem set. In addition to $\nu$-SDDP-fast and SDDP-mean, we plot the solution quality and its runtime obtained after different number of iterations of SDDP (denoted as SDDP-n with $n$ iterations). We observe that for the small-scale problem domain, SDDP-mean runs the fastest but with very high variance over the performance. For the large-scale problem domain, $\nu$-SDDP-fast achieves almost the same runtime as SDDP-mean (which roughly equals to the time for one round of SDDP forward pass). Also for large instances, SDDP would need to spend 1 or 2 magnitudes of runtime to match the performance of $\nu$-SDDP-fast. If we leverage $\nu$-SDDP-accurate to further update the solution in each test problem instance for just 10 iterations, we can further improve the solution quality. This suggests that our proposed $\nu$-SDDP achieves better time-solution trade-offs.

## 5.3 STUDY OF NUMBER OF GENERATED CUTTING PLANES

In Figure 3 we show the performance of $\nu$-SDDP-fast with respect to different model capacities, captured by the number of cutting planes the neural network can generate. A general trend indicates that more generated cutting planes would yield better solution quality. One exception lies in the Mid-Lng setting, where increasing the number of cutting planes beyond 64 would yield worse results. As we use the cutting planes generated by last $n$ iterations of SDDP solving in training $\nu$-SDDP-fast, our hypothesis is that the cutting planes generated by SDDP during the early stages in large problem settings would be of high variance and low-quality, which in turn provides noisy supervision. A more careful cutting plane pruning during the supervised learning stage would help resolve the problem.

## 5.4 LOW-DIMENSION PROJECT PERFORMANCE

Finally in Figure 4 we show the performance using low-rank projection. We believe that in reality customers from the same cluster (e.g., region/job based) would express similar behaviors, thus we created another synthetic environment where the customers form 4 clusters with equal size and thus have the same demand/transportation cost within each cluster. We can see that as long as the dimension goes above 80, our approach can automatically learn the low-dimension structure, and achieve much better performance than the baseline SDDP-mean. Given that the original decision problem is in 310-dimensional space, we expect having 310/4 dimensions would be enough, where the experimental results verified our hypothesis. We also show the low-dimension projection results for the problems with full-rank structure in Appendix E.1.

ACKNOWLEDGMENTS

The authors would like to thank Sherry Yang, Bethany Wang, Ben Sprecher and others from Cloud AI optimization, and the anonymous reviewers for their valuable feedbacks.

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

# Appendix

## A  MORE RELATED WORK

To scale up the MSSO solvers, a variety of hand-designed approximation schemes have been investigated. One natural approach is restricting the size of the scenario tree using either a scenario-wise or state-wise simplification. For example, as a scenario-wise approach, the expected value of perfect information (EVPI, (Birge, 1982; Hentenryck & Bent, 2006)) has been investigated for optimizing decision sequences within a scenario, which are then heuristically combined to form a full solution. Bareilles et al. (2020) instantiates EVPI by considering randomly selected scenarios in a progressive hedging algorithm (Rockafellar & Wets, 1991) with consensus combination. For a stage-wise approach, a two-stage model can be used as a surrogate, leveraging a bound on the approximation gap (Huang & Ahmed, 2009). All of these approximations rely on a fixed prior design for the reduction mechanism, and cannot adapt to a particular distribution of problem instances. Consequently, we do not expect such methods to be competitive with learning approaches that can adapt the approximation strategy to a given problem distribution.

**Difference to Learning from Cuts in Avila et al. (2021):**   the Batch Learning-SDDP (BL-SDDP) is released recently, where machine learning technique is also used for accelerating MSSO solver. However, this work is significant different from the proposed $\nu$-SDDP:

- Firstly and most importantly, the setting and target of these works are orthogonal: the BL-SDDP speeds up the SDDP for a particular given MSSO problem via parallel computation; while $\nu$-SDDP works for the meta-learning setting that learns from a dataset composed by plenty of MSSO problems sampled from a distribution, and the learning target is to generalize to new MSSO instances from the same distribution well;

- The technique contribution in BL-SDDP and $\nu$-SDDP are different. Specifically, BL-SDDP exploits existing off-policy RL tricks for accelerating the SDDP $Q$-update; while we proposed two key techniques for quick initialization **i)**, with predicted convex functions; **ii)**, dimension reduction techniques, to generalize different MSSOs and alleviate curse-of-dimension issues in SDDP, which has not been explored in BL-SDDP.

Despite being orthogonal, we think that the BL-SDDP can be used in our framework to provide better supervision for our cut function prediction, and serve as an alternative for fine-tuning after $\nu$-SDDP-fast.

One potential drawback of our $\nu$-SDDP is that, when the test instance distributions deviate a lot from what has been trained on, the neural initialization may predict cutting planes that are far away from the good ones, which may slow down the convergence of SDDP learning. Characterizing in-distribution v.s. out-of-distribution generalization and building the confidence measure is an important future work of current approach.

## B  PRACTICAL PROBLEM INSTANTIATION

In this section, we reformulate the inventory control and portfolio management as multi-stage stochastic decision problems.

### B.1  INVENTORY CONTROL

Let $S, V, C$ be the number of suppliers, inventories, and customers, respectively. We denote the parameters of the inventory control optimization as:

- procurement price matrix: $\mathbf{p}_t \in \mathbb{R}^{SV \times 1}$;
- sales price matrix: $\mathbf{q}_t \in \mathbb{R}^{VC \times 1}$;
- unit holding cost vector: $\mathbf{h}_t \in \mathbb{R}^{V \times 1}$;
- demand vector: $\mathbf{d}_t \in \mathbb{R}^{C \times 1}$;

- supplier capacity vector: $\mathbf{u}_t \in \mathbb{R}^{S \times 1}$;
- inventory capacity vector: $\mathbf{v}_t \in \mathbb{R}^{V \times 1}$;
- initial inventory vector $\mathbf{w}_0 \in \mathbb{R}^{V \times 1}$.

The decision variables of the inventory control optimization are denoted as:

- sales variable: $\mathbf{y}_t \in \mathbb{R}^{VC \times 1}$, indicating the amount of sales from inventories to customers at the beginning of stage $t$;
- procurement variable: $\mathbf{z}_t \in \mathbb{R}^{SV \times 1}$, indicating the amount of procurement at the beginning of stage $t$ after sales;
- inventory variable: $\mathbf{w}_t \in \mathbb{R}^{V \times 1}$, indicating the inventory level at the beginning of stage $t$ after procurement.

We denote the decision variables as

$$x_t = [\mathbf{y}_t, \mathbf{z}_t, \mathbf{w}_t],$$

the state as

$$\xi_t = [\mathbf{p}_t, \mathbf{q}_t, \mathbf{h}_t, \mathbf{d}_t, \mathbf{u}_t, \mathbf{v}_t, \mathbf{w}_0].$$

The goal of inventory management is to maximize the net profit for each stage, *i.e.*,

$$c_t^\top x_t := \mathbf{p}_t^\top \mathbf{z}_t + \mathbf{h}_t^\top \mathbf{w}_t - \mathbf{q}_t^\top \mathbf{y}_t,$$

and subject to the constraints of **1)** supplier capacity; **2)** inventory capacity; **3)** customer demand, *i.e.*,

$$\chi_t\left(x_{t-1}, \xi_t\right) = \left\{ \mathbf{y}_t, \mathbf{z}_t, \mathbf{w}_t \middle| \sum_{v=1}^{V} \mathbf{y}_t^v \leqslant \mathbf{d}_t \quad \text{(demand bound constraints)}, \right. \tag{12}$$

$$\sum_{v=1}^{V} \mathbf{z}_t^v \leqslant \mathbf{u}_t \quad \text{(supplier capacity constraints)} \tag{13}$$

$$\mathbf{w}_t \leqslant \mathbf{v}_t \quad \text{(inventory capacity constraints)} \tag{14}$$

$$\sum_{c=1}^{C} \mathbf{y}_t^c - \mathbf{w}_{t-1} \leqslant 0 \quad \text{(sales bounded by inventory)} \tag{15}$$

$$\sum_{s=1}^{S} \mathbf{z}_t^s - \sum_{c=1}^{C} \mathbf{y}_t^c + \mathbf{w}_{t-1} = \mathbf{w}_t \quad \text{(inventory transition)} \tag{16}$$

$$\left. \mathbf{z}_t, \mathbf{y}_t, \mathbf{w}_t, \geqslant 0 \quad \text{(non-negativity constraints)} \right\}. \tag{17}$$

To sum up, the optimization problem can be defined recursively as follows:

$$\min_{x_1} \quad c_1^\top x_1 + \mathbb{E}_{\xi_2} \left[ \min_{x_2} c_2\left(\xi_2\right)^\top x_2\left(\xi_2\right) + \mathbb{E}_{\xi_3} \left[ \cdots + \mathbb{E}_{\xi_T} \left[ \min_{x_T} c_T\left(\xi_T\right)^\top x_T\left(\xi_T\right) \right] \cdots \right] \right], \tag{18}$$

$$\text{s.t.} \quad x_t \in \chi_t(x_{t-1}, \xi_t), \ \forall t = \{1, \ldots, T\}. \tag{19}$$

In fact, the inventory control problem (18) is simplified the multi-stage stochastic decision problem (3) by considering state independent transition.

## B.2 Portfolio Management

Let $I$ be the number of assets, *e.g.*, stocks, being managed. We denote the parameters of the portfolio optimization are:

- ask (price to pay for buying) open price vector $\mathbf{p}_t \in \mathbb{R}^{I \times 1}$;
- bid (price to pay for sales) open price vector $\mathbf{q}_t \in \mathbb{R}^{I \times 1}$;
- initial amount of investment vector $\mathbf{w}_0 \in \mathbb{R}^{I \times 1}$;
- initial amount of cash $r_0 \in \mathbb{R}_+$.

The decision variables of the portfolio optimization are:

- sales vector $\mathbf{y}_t \in \mathbb{R}^{I \times 1}$, indicating the amount of sales of asset $i$ at the beginning of stage $t$.

- purchase vector $\mathbf{z}_t \in \mathbb{R}^{I \times 1}$, indicating the amount of procurement at the beginning of stage $t$;

- holding vector $\mathbf{w}_t \in \mathbb{R}^{I \times 1}$, indicating the amount of assets at the beginning of stage $t$ after purchase and sales;

- cash scalar $r_t$, indicating the amount of cash at the beginning of stage $t$.

We denote the decision variables as

$$x_t = [\mathbf{y}_t, \mathbf{z}_t, \mathbf{w}_t, r_t],$$

and the state as

$$\xi_t = [\mathbf{p}_t, \mathbf{q}_t].$$

The goal of portfolio optimization is to maximize the net profit *i.e.*,

$$c_t^\top x_t := \mathbf{p}_t^\top \mathbf{z}_t - \mathbf{q}_t^\top \mathbf{y}_t,$$

subject to the constraints of initial investment and the market prices, *i.e.*,

$$\chi_t\left(x_{t-1}, \xi_t\right) := \left\{ \mathbf{y}_t, \mathbf{z}_t, \mathbf{w}_t, r_t \; \middle| \; \mathbf{y}_t - \mathbf{w}_{t-1} \leqslant 0 \quad \text{(individual stock sales constraints)} \right. \tag{20}$$

$$\mathbf{p}_t^\top \mathbf{z}_t - r_{t-1} \leqslant 0 \quad \text{(stock purchase constraints)} \tag{21}$$

$$\mathbf{y}_t - \mathbf{z}_t + \mathbf{w}_t - r_{t-1} = 0 \tag{22}$$

$$\text{(individual stock position transition)}$$

$$\mathbf{q}_t^\top \mathbf{y}_t - \mathbf{p}_t^\top \mathbf{z}_t - r_t + r_{t-1} = 0 \tag{23}$$

$$\left. \text{(cash position transition)} \right\}.$$

With the $c_t$ and $\chi_t\left(x_{t-1}, \xi_t\right)$ defined above, we initiate the multi-stage stochastic decision problem (3) for portfolio management.

## C  DETAILS ON STOCHASTIC DUAL DYNAMIC PROGRAMMING

We have introduced the SDDP in Section 2. In this section, we provide the derivation of the updates in forward and backward pass,

- **Forward pass**, updating the action according to (5) based on the current estimation of the value function at each stage via (4). Specifically, for $i$-th iteration of $t$-stage with sample $\xi_t^j$, we solve the optimization

$$x_t \in \operatorname*{argmin}_{x_t \in \chi_t\left(x_{t-1}, \xi_t^j\right)} c_t\left(\xi_t^j\right)^\top x_t + V_{t+1}^i\left(x_t\right). \tag{24}$$

In fact, the $V$-function is a convex piece-wise function. Specifically, for $i$-th iteration of $t$-stage, we have

$$V_{t+1}^i\left(x_t\right) = \max_{k \leqslant i} \left\{ \left(\beta_{t+1}^k\right)^\top x_t + \alpha_{t+1}^k \right\},$$

Then, we can rewrite the optimization (24) into standard linear programming, *i.e.*,

$$\min_{x_t, \theta_{t+1}} \quad c_t\left(\xi_t^j\right)^\top x_t + \theta_{t+1} \tag{25}$$

$$\text{s.t.} \quad A_t\left(\xi_t^j\right) x_t = b_t\left(\xi_t^j\right) - B_{t-1}\left(\xi_t^j\right) x_{t-1}, \tag{26}$$

$$- \left(\beta_{t+1}^k\right)^\top x_t + \theta_{t+1} \geqslant \alpha_{t+1}^k, \forall k = 1, \ldots, i, \tag{27}$$

$$x_t \geqslant 0, \tag{28}$$

- **Backward pass**, updating the estimation of the value function via the dual of (25), *i.e.*, for $i$-th iteration of $t$-stage with sample $\xi_t^j$, we calculate

$$\max_{\omega_t, \rho_t} \quad \left( b_t\left(\xi_t^j\right) - B_{t-1}\left(\xi_t^j\right) \right)^\top \omega_t + \sum_{k=1}^{i} \rho_t^k {\alpha^k}_{t+1}, \tag{29}$$

$$\text{s.t.} \quad A_t\left(\xi_t^j\right)^\top \omega_t - \sum_{k=1}^{i} \rho_t^k \left(\beta_{t+1}^k\right)^\top \leqslant c_t\left(\xi_t^j\right), \tag{30}$$

$$-1 \leqslant \rho_t^\top \mathbf{1} \leqslant 1. \tag{31}$$

Then, we have the

$$V_t^{i+1}\left(x_{t-1}\right) = \max\left\{ V_t^i\left(x_{t-1}\right), v_t^{i+1}\left(x_{t-1}\right) \right\}, \tag{32}$$

which is still convex piece-wise linear function, with

$$v_t^{i+1}\left(x_{t-1}\right) := \left(\beta_t^{i+1}\right)^\top x_{t-1} + \alpha_t^{i+1}, \tag{33}$$

where

$$\left(\beta_t^{i+1}\right)^\top := \frac{1}{m} \sum_{j=1}^{m} \left[ -B_{t-1}\left(\xi_t^j\right)^\top \omega_t\left(\xi_t^j\right) \right],$$

$$\alpha_t^{i+1} := \frac{1}{m} \sum_{j=1}^{m} \left[ b_t\left(\xi_t^j\right)^\top \omega_t\left(\xi_t^j\right) + \sum_{k=1}^{i} \alpha_{t+1}^k\left(\xi_t^j\right) \rho_t^k\left(\xi_t^j\right) \right],$$

with $\left(\omega_t\left(\xi_t\right), \rho_t\left(\xi_t\right)\right)$ as the optimal dual solution with realization $\xi_t$.

In fact, although we split the forward and backward pass, in our implementation, we exploit the primal-dual method for LP, which provides both optimal primal and dual variables, saving the computation cost.

Note that SDDP can be interpreted as a form of TD-learning using a non-parametric piecewise linear model for the $V$-function. It exploits the property induced by the parametrization of value functions, leading to the update w.r.t. $V$-function via adding dual component by exploiting the piecewise linear structure in a closed-form functional update. That is, TD-learning (Sutton & Barto, 2018; Bertsekas, 2001) essentially conducts stochastic approximate dynamic programming based on the Bellman recursion (Bellman, 1957).

## D  NEURAL NETWORK AND LEARNING SYSTEM DESIGN

**Neural network design:**  In Figure 5 we present the design of the neural network that tries to approximate the $V$-function. The neural network takes two components as input, namely the feature vector that represents the problem configuration, and the integer that represents the current stage of the multi-stage solving process. The stage index is converted into 'time-encoding', which is a 128-dimensional learnable vector. We use the parameters of distributions as the feature of $\phi\left(P(\xi)\right)$ for simplicity. The characteristic function or kernel embedding of distribution can be also used here. The feature vector will also be projected to the same dimension and added together with the time encoding to form as the input to the MLP. The output of MLP is a matrix of size $k \times (N + 1)$, where $k$ is the number of linear pieces, $N$ is the number of variable (and we also need one additional dimension for intercept). The result is a piecewise linear function that specifies a convex lowerbound. We show an illustration of 2D case on the right part of the Figure 5.

This neural network architecture is expected to adapt to different problem configurations and time steps, so as to have the generalization and transferability ability across different problem configurations.

**Remark (Stochastic gradient computation):**  Aside from $G_\psi$, unbiased gradient estimates for all other variables in the loss (9) can be recovered straightforwardly. However, $G_\psi$ requires special treatment since we would like it to satisfy the constraints $G_\psi^\top G_\psi = I_p$. Penalty method is one of the choices, which switches the constraints to a penalty in objective, *i.e.*, $\eta \left\| G_\psi^\top G_\psi - I_p \right\|_2^2$. However, the solution satisfies the constraints, only if $\eta \to \infty$ (Nocedal & Wright, 2006, Chapter 17). We

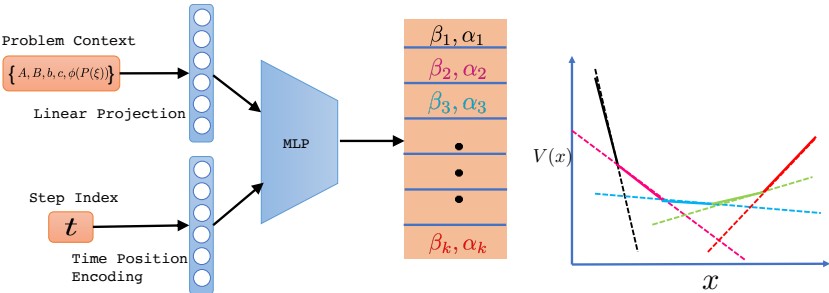

Figure 5: Hypernet style parameterization of neural $V$-function.

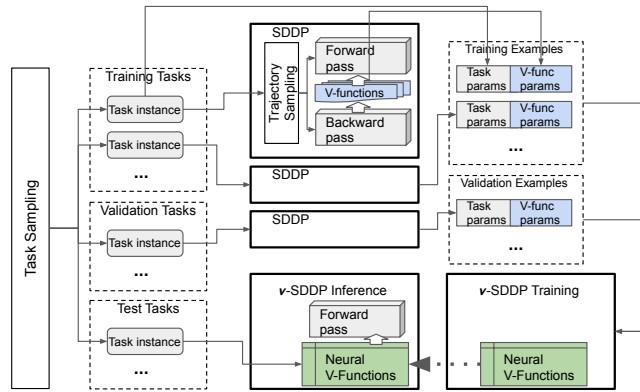

Figure 6: Illustration of the overall system design.

derive the gradient over the Stiefel manifold (Mahony et al., 1996), which ensures the orthonormal constraints,

$$\text{grad}_{G_\psi}\ell = \left(I - G_\psi^\top G_\psi\right)\Xi G_\psi^\top, \tag{34}$$

with $\Xi := \sum_t \sum_{i=1}^n \sum_j^m x_{tj}^{i*}\left(x_{tj}^{i*}\right)^\top$. Note that this gradient can be estimated stochastically since $\Xi$ can be recognized as an expectation over samples.

The gradients on Stiefel manifold $\left\{G|G^\top G = I\right\}$ can be found in Mahony et al. (1996). We derive the gradient (34) via Lagrangian for self-completeness, following Xie et al. (2015).

Consider the Lagrangian as

$$L\left(G_\psi, \Lambda\right) = \sum_{z \in \mathcal{D}_n} \ell\left(W; z\right) + \text{tr}\left(\left(G_\psi^\top G_\psi - I\right)\Lambda\right),$$

where the $\Lambda$ is the Lagrangian multiplier. Then, the gradient of the Lagrangian w.r.t. $G_\psi$ is

$$\nabla_{G_\psi}L = 2\Xi G_\psi^\top + G_\psi^\top\left(\Lambda + \Lambda^\top\right). \tag{35}$$

With the optimality condition

$$\nabla_{G_\psi, \Lambda}L = 0 \Rightarrow \begin{cases} G_\psi^\top G_\psi - I = 0 \\ 2\Xi G_\psi^\top + G_\psi^\top\left(\Lambda + \Lambda^\top\right) = 0 \end{cases} \Rightarrow -2G_\psi\Xi G_\psi^\top = \left(\Lambda + \Lambda^\top\right). \tag{36}$$

Plug (36) into the gradient (35), we have the optimality condition,

$$\underbrace{\left(I - G_\psi^\top G_\psi\right)\Xi G_\psi^\top}_{\text{grad}_{G_\psi}} = 0. \tag{37}$$

To better numerical isolation of the individual eigenvectors, we can exploit Gram-Schmidt process into the gradient estimator (34), which leads to the generalized Hebbian rule (Sanger, 1989; Kim et al., 2005; Xie et al., 2015),

$$\widetilde{\text{grad}}_{G_\psi}\ell = \left(I - \mathcal{LT}\left(G_\psi^\top G_\psi\right)\right)\Xi G_\psi^\top = \Xi G_\psi^\top - \mathcal{LT}\left(G_\psi^\top G_\psi\right)\Xi G_\psi^\top. \tag{38}$$

The $\mathcal{LT}\left(\cdot\right)$ extracts the lower triangular part of a matrix, setting the upper triangular part and diagonal to zero, therefore, is mimicking the Gram-Schmidt process to subtracts the contributions from each

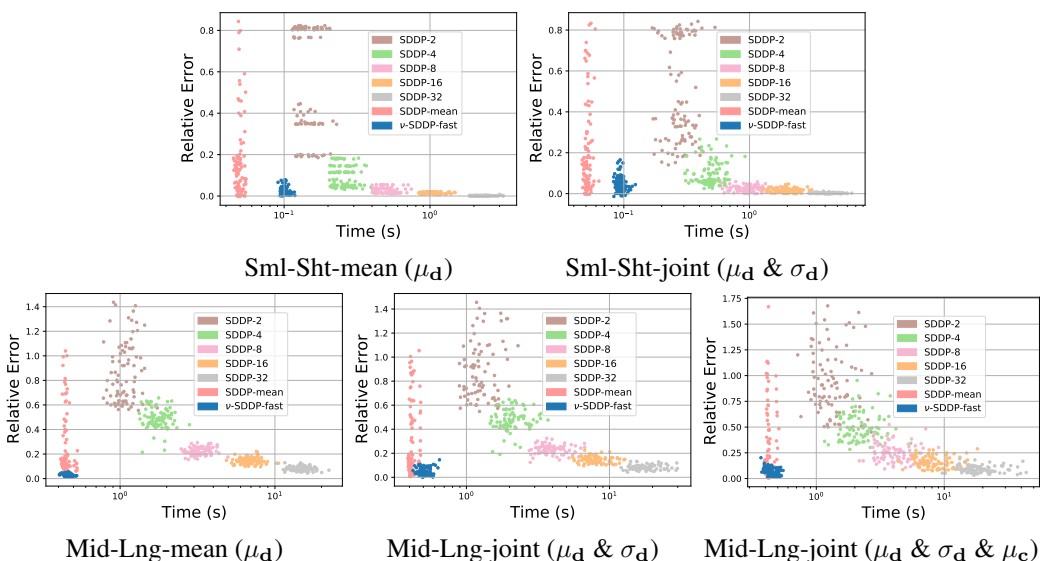

Figure 7: Time-solution trade-off.

other eigenvectors to achieve orthonormality. Sanger (1989) shows that the updates with (38) will converges to the first $p$ eigenvectors of $\Xi$.

**System design:**  Next we present the entire system end-to-end in Figure 6.

- *Task sampling*: the task sampling component draws the tasks from the same *meta* distribution. Note that each task is a specification of the distribution (*e.g.*, the center of Gaussian distribution), where the specification follows the same *meta* distribution.

- We split the task instances into train, validation and test splits:
    - *Train:* We solve each task instance using SDDP. During the solving of SDDP we need to perform multiple rounds of forward pass and backward pass to update the cutting planes ($V$-functions), as well as sampling trajectories for monte-carlo approximation. The learned neural $V$-function will be used as initialization. After SDDP solving converges, we collect the corresponding task instance specification (parameters) and the resulting cutting planes at each stage to serve as the training supervision for our neural network module.
    - *Validation:* We do the same thing for validation tasks, and during training of neural network we will dump the models that have the best validation loss.
    - *Test:* In the test stage, we also solve the SDDP until convergence as groundtruth, which is only used for evaluating the quality of different algorithms. For our neural network approach, we can generate the convex lower-bound using the trained neural nework, conditioning on each pair of (test task instance specification, stage index). With the predicted $V$-functions, we can run the forward pass only once to retrieve the solution at each stage. Finally we can evaluate the quality of the obtained solution with respect to the optimal ones obtained by SDDP.

# E  MORE EXPERIMENTS

## E.1  INVENTORY OPTIMIZATION

### E.1.1  ADDITIONAL RESULTS ON $\nu$-SDDP

We first show the full results of time-solution quality trade-off in Figure 7, and how $\nu$-SDDP-accurate improves from $\nu$-SDDP-fast with better trade-off than SDDP solver iterations in Figure 8. We can see the conclution holds for all the settings, where our proposed $\nu$-SDDP achieves better trade-off.

Then we also show the ablation results of using different number of predicted cutting planes in Figure 9. We can see in all settings, generally the more the cutting planes the better the results. This suggests that in higher dimensional case it might be harder to obtain high quality cutting planes, and

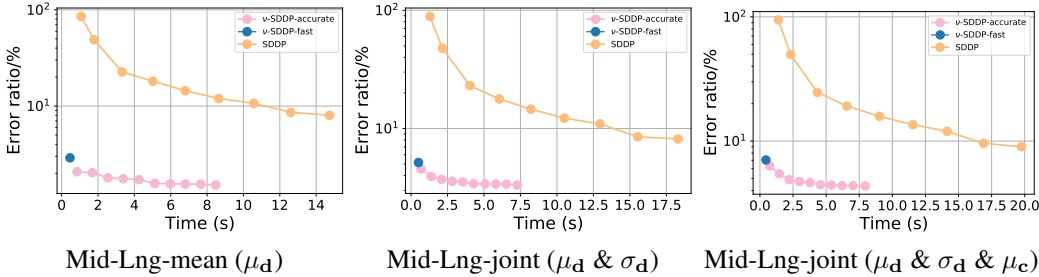

Figure 8: Time-solution trade-off when $\nu$-SDDP-accurate improves the solution from $\nu$-SDDP-fast further.

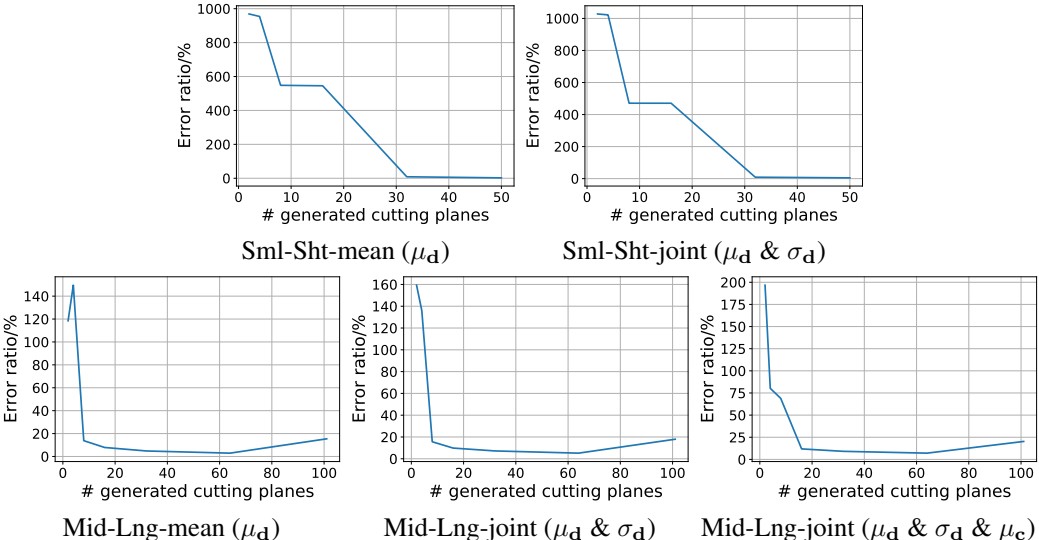

Figure 9: Ablation: number of generated cutting planes.

due to the convex-lowerbound nature of the $V$-function, having a bad cutting plane could possibly hurt the overall quality. How to prune and prioritize the cutting planes will be an important direction for future works.

We provide the full ablation results of doing low-dimensional projection for solving SDDP in Figure 10. The trend generally agrees with our expectation that, there is a trade-off of the low-dimensionality that would balance the quality of LP solving and the difficulty of neural network learning.

**Longer horizon:** we further experiment with the Mid-Lng-joint ($\mu_d \& \sigma_d \& \mu_c$) by varying $T$ in $\{10, 20, 30, 40, 50\}$. See Table 4 for more information.

Table 4: Average error ratio of $\nu$-SDDP-fast on Mid-Lng-joint ($\mu_d \& \sigma_d \& \mu_c$) setting with varying $T$.

| Horizon length | 10 | 20 | 30 | 40 | 50 |
|---|---|---|---|---|---|
| Average error ratio | 3.29% | 3.47% | 3.53% | 2.65% | 0.82% |

### E.1.2 ADDITIONAL RESULTS ON MODEL-FREE RL ALGORITHMS

We implemented the inventory control problem as an environment in the Tensorflow TF-Agents library and used the implementation of DQN (Mnih et al., 2013)[1], DDPG (Lillicrap et al., 2016), PPO (Schulman et al., 2017) and SAC (Haarnoja et al., 2018) from the TF-Agent to evaluate the performance of these four model-free RL algorithms. Note that the TF-Agent environment follows a

---

[1]We provided a simple extension of DQN to support multi-dimensional actions.

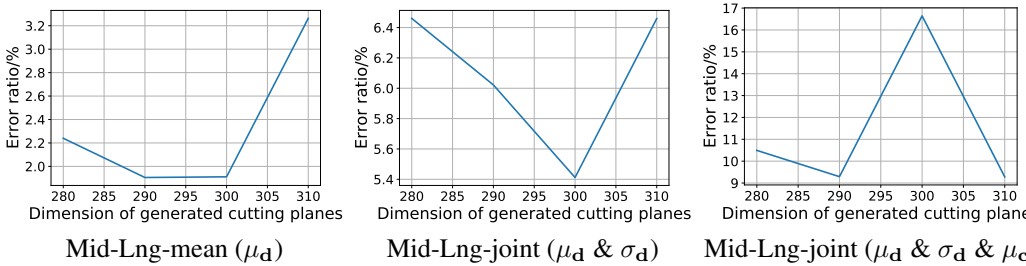

Mid-Lng-mean ($\mu_{\mathbf{d}}$)     Mid-Lng-joint ($\mu_{\mathbf{d}}$ & $\sigma_{\mathbf{d}}$)     Mid-Lng-joint ($\mu_{\mathbf{d}}$ & $\sigma_{\mathbf{d}}$ & $\mu_{\mathbf{c}}$)

Figure 10: Low-dim projection results when the underlying problem *does not* have a low-rank structure.

| Task | Parameter Domain | DQN | DDPG | PPO | SAC |
|------|------------------|-----|------|-----|-----|
| Sml-Sht | demand mean ($\mu_{\mathbf{d}}$) | $1157.86 \pm 452.50\%$ | $28.62 \pm 8.69\,\%$ | $2849.931 \pm 829.91\%$ | $38.42 \pm 17.78\%$ |
| | joint ($\mu_{\mathbf{d}}$ & $\sigma_{\mathbf{d}}$) | $3609.62 \pm 912.54\%$ | $100.00 \pm 0.00\,\%$ | $3273.71 \pm 953.13\%$ | $33.08 \pm 8.05\%$ |
| Mid-Long | demand mean ($\mu_{\mathbf{d}}$) | $5414.15 \pm 1476.21\%$ | $100.00 \pm 0.00\%$ | $5411.16 \pm 1474.19\%$ | $17.81 \pm 10.26\%$ |
| | joint ($\mu_{\mathbf{d}}$ & $\sigma_{\mathbf{d}}$) | $5739.68 \pm 1584.63\%$ | $100.00 \pm 0.00\%$ | $5734.75 \pm 1582.68\,\%$ | $50.19 \pm 5.57\%$ |
| | joint ($\mu_{\mathbf{d}}$ & $\sigma_{\mathbf{d}}$ & $\mu_{\mathbf{c}}$) | $6382.87 \pm 2553.527\%$ | $100.00 \pm 0.00\%$ | $6377.93 \pm 2550.03\,\%$ | $135.78 \pm 17.12\%$ |

Table 6: Average Error Ratio of Objective Value.

MDP formulation. When adapting to the inventory control problem, the states of the environment are the levels of the inventories and the actions are the amount of procurement and sales. As a comparison, the states and actions in the MDP formulation collectively form the decision variables in the MSSO formulation and their relationship – inventory level transition, is captured as a linear constraint. Following Schulman (2016), we included the timestep into state in these RL algorithms to have non-stationary policies for finite-horizon problems. In the experiments, input parameters for each problem domain and instances are normalized for model training for policy gradient algorithms and the results are scaled back for reporting.

We report the average error ratio of these four RL algorithms in Table. 6 along with the average variance in Table. 7. Note that the SAC performance is also reported in the main text in Table. 2 and Table. 3. The model used in the evaluation is selected based on the best mean return over the 50 trajectories from the validation environment, based on which the hyperparameters are also tuned. We report the selected hyperparameters for each algorithm in Table. 5. We use MLP with 3 layers as the $Q$-network for DQN, as the actor network and the critic/value network for SAC, PPO and DDPG. All networks have the same learning rate and with a dropout parameter as 0.001.

Table 5: Hyperparameter Selections.

| Algorithm | Hyperparameters |
|-----------|-----------------|
| SAC | learning rate(0.01), num MLP units (50), target update period (5), target update tau (0.5) |
| PPO | learning rate(0.001), num MLP units (50), target update period (5), target update tau (0.5) |
| DQN | learning rate(0.01), num MLP units (100), target update period (5), target update tau (0.5) |
| DDPG | learning rate(0.001), num MLP units (50), ou stddev (0.2), ou damping (0.15) |

We see that SAC performs the best among the four algorithms in terms of solution quality. All the algorithms can not scale to Mid-Long setting. DDPG, for example, produces a trivial policy of no action in most of setups (thus has an error ratio of 100). The policies learned by DQN and PPO are even worse, producing negative returns[2].

To understand the behavior of each RL algorithm, we plotted the convergence of the average mean returns in Figure. 11 for the Sml-Sht task. In each plot, we show four runs of the respective algorithm under the selected hparameter. We could see that though SAC converges the slowest, it is able to

---

[2]Negative returns are caused by over-procurement in the early stages and the leftover inventory at the last stage.

| Task | Parameter Domain | DQN | DDPG | PPO | SAC |
|---|---|---|---|---|---|
| Sml-Sht | demand mean ($\mu_\mathbf{d}$) | $46.32 \pm 85.90$ | $0.34 \pm 0.22$ | $119.08 \pm 112.00$ | $3.90 \pm 8.39$ |
| | joint ($\mu_\mathbf{d}$ & $\sigma_\mathbf{d}$) | $86.097 \pm 100.81$ | $0.00 \pm 0.00$ | $169.08 \pm 147.24$ | $1.183 \pm 4.251$ |
| Mid-Long | demand mean ($\mu_\mathbf{d}$) | $1334.30 \pm 270.00$ | $0.00 \pm 0.00$ | $339.97 \pm 620.01$ | $1.98 \pm 2.65$ |
| | joint ($\mu_\mathbf{d}$ & $\sigma_\mathbf{d}$) | $1983.71 \pm 1874.61$ | $0.00 \pm 0.00$ | $461.27 \pm 1323.24$ | $205.51 \pm 150.90$ |
| | joint ($\mu_\mathbf{d}$ & $\sigma_\mathbf{d}$ & $\mu_\mathbf{c}$) | $1983.74 \pm 1874.65$ | $0.00 \pm 0.00$ | $462.74 \pm 1332.30$ | $563.19 \pm 114.03$ |

Table 7: Objective Value Variance.

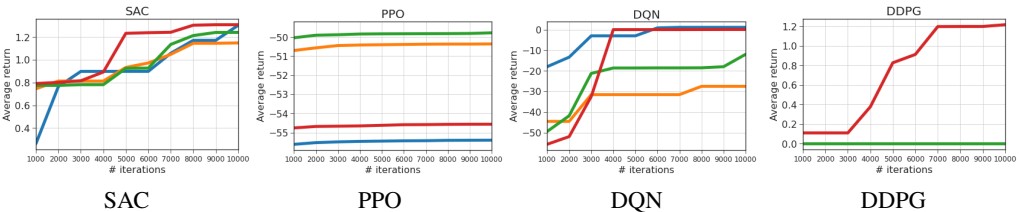

Figure 11: Average mean return (values are normalized with optimal mean value as $1.736$).

achieve the best return. For all algorithms, their performance is very sensitive to initialization. DDPG, for example, has three runs with 0 return, while one run with a return of 1.2. For PPO and DQN, the average mean returns are both negative .

We further check the performance of these algorithms in the validation environment based on the same problem instance (*i.e.*, the same problem parameters) as in SDDP-mean, where the model is trained and selected. We expect this would give the performance upper bound for these algorithms. Again similar results are observed. The best return mean over the validation environment is $-38.51$ for PPO, $0.95$ for DQN, $1.31$ for SAC and $1.41$ for DDPG, while the SDDP optimal return value is $1.736$. It is also worth noting that DDPG shows the strongest sensitivity to initialization and its performance drops quickly when the problem domain scales up.

## E.2 PORTFOLIO OPTIMIZATION

We use the daily opening prices from selected Nasdaq stocks in our case study. This implies that the asset allocation and rebalancing in our portfolio optimization is performed once at each stock market opening day. We first learn a probabilistic forecasting model from the historical prices ranging from 2015-1-1 to 2020-01-01. Then the forecasted trajectories are sampled from the model for the stochastic optimization. Since the ask price is always slightly higher than the bid price, at most one of the buying or selling operation will be performed for each stock, but not both, on a given day.

### E.2.1 STOCK PRICE FORECAST

The original model for portfolio management in Appendix B.2 is too restrict. We generalize the model with autoregressive process (AR) of order $o$ with independent noise is used to model and predict the stock price:

$$\mathbf{p}_t = \sum_{i=1}^{o}(\phi_i \mathbf{p}_{t-i} + \epsilon_i^r) + \epsilon^o$$

where $\phi_i$ is the autoregressive coefficient and $\epsilon^r_i \sim \mathcal{N}(0, \sigma_i^2)$ is a white noise of order $i$. $\epsilon^o \sim \mathcal{N}(0, \sigma_o^2)$ is a white noise of the observation. Each noise term is $\epsilon$ assumed to be independent. It is easy to check that the MSSO formulation for portfolio management is still valid by replacing the expectation for $\epsilon$, and setting the concatenate state $[\mathbf{p}_t^{t-o}, \mathbf{q}_t]$, where $\mathbf{p}_t^{t-o} := [\mathbf{p}_{t-i}]_{i=0}^{o}$.

We use variational inference to fit the model. A variational loss function (i.e., the negative evidence lower bound (ELBO)) is minimized to fit the approximate posterior distributions for the above parameters. Then we use the posterior samples as inputs for forecasting. In our study, we have studied the forecasting performance with different length of history, different orders of the AR process and different groups of stocks. Figure. 12 shows the ELBO loss convergence behavior under different setups. As we can see, AR models with lower orders converge faster and smoother.

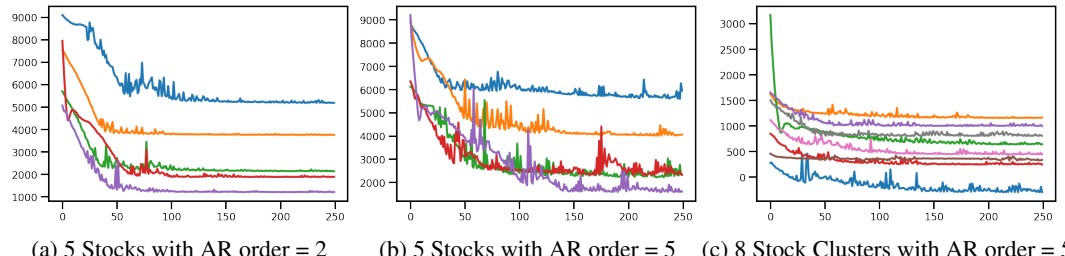

(a) 5 Stocks with AR order = 2      (b) 5 Stocks with AR order = 5      (c) 8 Stock Clusters with AR order = 5

Figure 12: Evidence lower bound (ELBO) loss curve.

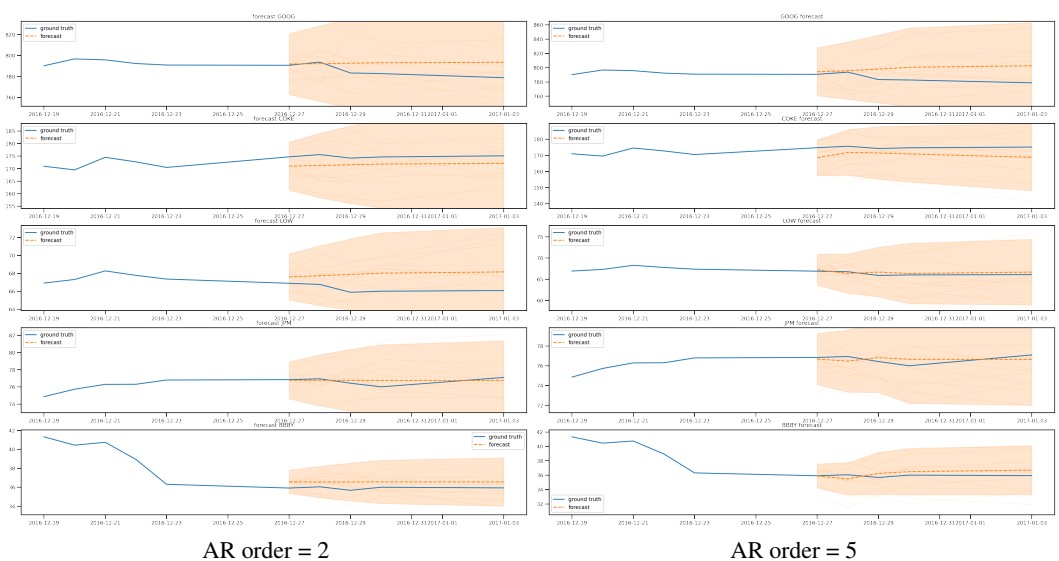

AR order = 2                      AR order = 5

Figure 13: Probablistic Forecast of 5 Stocks with Different AR Orders.

We further compare the forecasting performance with different AR orders. Figure. 13 plots a side-by-side comparison of the forecasted mean trajectory with a confidence interval of two standard deviations (95%) for 5 randomly selected stocks (with tickers GOOG, COKE, LOW, JPM, BBBY) with AR order of 2 and 5 from 2016-12-27 for 5 trading days. As we could see, a higher AR order (5) provides more time-variation in forecast and closer match to the ground truth.

In addition, we cluster the stocks based on their price time series (i.e., each stock is represented by a $T$-dimensional vector in the clustering algorithm, where $T$ is the number of days in the study). We randomly selected 1000 stocks from Nasdaq which are active from 2019-1-1 to 2020-1-1 and performed $k$-means clustering to form 8 clusters of stocks. We take the cluster center time series as the training input. Figure. 12(c) shows the ELBO loss convergence of an AR process of order 5 based on these 8 cluster center time series. As we see, the stock cluster time series converge smoother and faster compared with the individual stocks as the aggregated stock series are less fluctuated. The forecasting trajectories of these 8 stock clusters starting from 2019-03-14 are plotted in Figure. 14.

### E.2.2 SOLUTION QUALITY COMPARISON

Table 8: Portfolio optimization with synthetic standard deviation of stock price.

|  | SDDP-mean | $\nu$-SDDP-zeroshot |
|---|---|---|
| 5 stocks (STD scaled by 0.01) | $290.05 \pm 221.92$ % | $\mathbf{1.72 \pm 4.39}$ % |
| 5 stocks (STD scaled by 0.001) | $271.65 \pm 221.13$ % | $\mathbf{1.84 \pm 3.67}$ % |
| 8 clusters (STD scaled by 0.1) | $69.18 \pm 77.47$ % | $\mathbf{1.43e^{-6} \pm 4.30e^{-5}}$ % |
| 8 clusters (STD scaled by 0.01) | $65.81 \pm 77.33$ % | $\mathbf{3.25e^{-6} \pm 3.44e^{-5}}$ % |

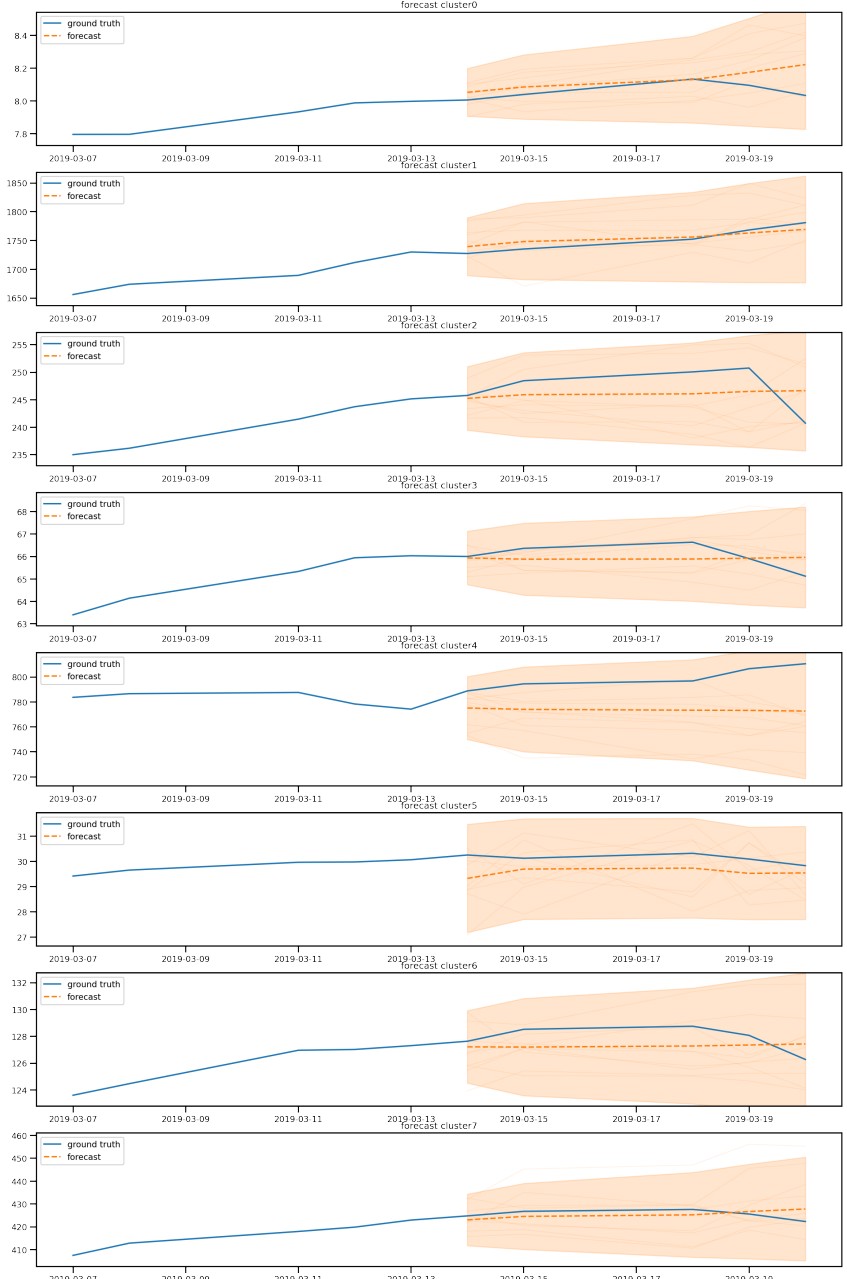

Figure 14: Probablistic Forecast of 8 Stock Clusters.

With an AR forecast model of order $o$, the problem context of a protfolio optimizaton instance is then captured by the joint distribution of the historical stock prices over a window of $o$ days. We learn this distribution using kernel density estimation. To sample the scenarios for each stage using the AR model during training for both SDDP and $\nu$-SDDP, we randomly select the observations from the previous stage to seed the observation sampling in the next stage. Also we approximate the state representation by dropping the term of $\mathbf{p}_t^{t-o}$. With such treatments, we can obtain SDDP results with manageable computation cost. We compare the performance of SDDP-optimal, SDDP-mean and $\nu$-SDDP-zeroshot under different forecasting models for a horizon of $5$ trading days. First we observe that using the AR model with order $2$ as the forecasting model, as suggested by the work (Dantzig & Infanger, 1993), produce a very simple forecasting distribution where the mean is monotonic over the forecasting horizon. As a result, all the algorithms will lead to a simple "buy and hold" policy which make no difference in solution quality. We further increase the AR order gradually

from 2 to 5 and find AR order at 5 produces sufficient time variation that better fits the ground truth. Second we observe that the variance learned from the real stock data based on variational inference is significant. With high variance, both SDDP-optimal and $\nu$-SDDP-zeroshot would achieve the similar result, which is obtained by a similar "buy and hold" policy. To make the task more challenging, we rescale the standard deviation (STD) of stock price by a factor in Table 8 for both the 5-stock case and 1000-stock case with 8 clusters situations. For the cluster case, the $\nu$-SDDP-zeroshot can achieve almost the same performance as the SDDP-optimal.

### E.3    COMPUTATION RESOURCE

For the SDDP algorithms we run using multi-core CPUs, where the LP solving can be parallelized at each stage. For RL based approaches and our $\nu$-SDDP, we train using a single V100 GPU for each hparameter configuration for at most 1 day or till convergence.

