# OpenReview forum: "Neural Stochastic Dual Dynamic Programming"
_ICLR.cc/2022/Conference — ICLR 2022 Poster_

### Official Review · Reviewer_RrLt · 2021-11-02

**Correctness:** 3
**Technical Novelty And Significance:** 3
**Empirical Novelty And Significance:** 2
**Recommendation:** 6
**Confidence:** 4

**Main Review:**

Overall, the paper is clear and investigates an intuitive extension of an SDDP framework with a learning mechanism. In particular, it is natural to consider more sophisticated value-function estimate procedures when the method is inherently iterative, such as in the case of SDDP and other cut-generation procedure methods. The numerical results, while still somewhat restricted to only two benchmarks, provide interesting insights on the trade-off between computational times and solution quality, as well as promising results.

However, I have two major concerns. First, this broader idea of learning from cuts is not new, even within the context of SDDPs. More specifically, I wonder what would be the connections and differences with respect to "batch learning" in SDDPs, which is similar in that it learns a "Q" function while traversing the state paths. Nonetheless, I believe that any other learning model could also be suitable. A recent reference is from EJOR:

http://www.optimization-online.org/DB_FILE/2021/05/8397.pdf

Second, the paper lacks a deeper, more thorough investigation of when the procedure is beneficial or not. It tends to oversell with somewhat bold statements (e.g., "v-SSDP reduces the curse of dimensionality effect," "gain a significant advantage over random initialization" etc), but it is not clear to me whether that should always be the case rigorously. Further, only two benchmarks are not sufficient to make such general statements.

For instance, what are the properties of the value function (in view of the piecewise elements) so that learning is effective? Could there be a case where the convergence of learning is so slow so that the procedure just adds an overhead? Why only random starts, and not something possibly more systematic but easier to implement (e.g., perhaps https://www.sciencedirect.com/science/article/pii/S0377042715002794?) It would be relevant to precisely identify, at least intuitive, problem structure where we can more easily grasp that random, or any more systematic procedure, could still be beneficial.




**Summary Of The Paper:**

The paper develops a stochastic dual dynamic programming (SDDP) approach that incorporates a value-function prediction component. Specifically, the authors use opportunistic forward and backward passes to train a neural network model that estimates the value function, which is subsequently used to initialize and guide the basis SDDP iteration procedure. The procedure also leverages the actual value functions SDDP updates to re-train the predictive model for further epochs. Numerical results on inventory and portfolio optimization suggest better error ratios in comparison to SDDP variants and a reinforcement learning model.


**Summary Of The Review:**

The paper develops an intuitive and natural idea, with promising results on two benchmarks. However, the paper (i) lacks connection with closely related literature (batch learning in SDDPs), and, in my view, (ii) it does not deeply investigate trade-offs of the methodology and underlying structural properties, besides a limited study on the cut generation procedure.

---

> ### Author Response · Authors · 2021-11-23
> **Reply to Reviewer RrLt's review (2/2)**
>
> ### **"Why only random starts, and not … e.g. perhaps [De Matos et.al](https://www.sciencedirect.com/science/article/pii/S0377042715002794)"**
>
> We thank the reviewer for pointing out this paper. We want to clarify that we are not randomly initializing the cutting planes as these will interfere with the lower-bounds. One way we explored is to initialize the cut with $y=\text{negative large number}$. We also explored another way, where the initialization of the SDDP cutting planes starts from a problem-specific random initial feasible solution (for example, in the inventory management, it is set to the initial-inventory specified by the problem definition) and then the backward pass from it.
>
> The paper you referred to studied the tree-traversal strategies and the cut selection methods, which doesn’t have an impact on initialization. Nevertheless, the advanced techniques of SDDP are not a conflict to our framework as we can seamlessly incorporate them during meta-learning training and adaptation fine tuning stages.

---

> > ### Comment · Reviewer_RrLt · 2021-11-26
> > **Acknowledgement**
> >
> > Thank you for clarifying my questions, I appreciate it.

---

> ### Author Response · Authors · 2021-11-23
> **Reply to Reviewer RrLt's review (1/2)**
>
> Dear reviewer, thanks for your valuable comments. Please see our response below:
>
> ### **Difference between Batch Learning SDDP  and $\nu$-SDDP**
>
> We would like to thank the reviewer for pointing out the relevant work. However, there are several key differences between our $\nu$-SDDP and the Batch-Learning SDDP (BL-SDDP).
> - Firstly and most importantly, the setting and target of these works are orthogonal: The BL-SDDP speeds up the SDDP for *a particular given MSSO* problem via parallel computation; while $\nu$-SDDP works for the *meta-learning* setting that learns from a dataset composed by *plenty of MSSO* problems sampled from a distribution, and the learning target is to *generalize* to new MSSO instances from the same distribution well.
> - The technique contribution in BL-SDDP and $\nu$-SDDP are different. Specifically, BL-SDDP exploits existing off-policy RL tricks for accelerating the SDDP Q-update; while we proposed two key techniques for quick initialization *i)*,  with predicted convex functions; *ii)*, dimension reduction techniques, to generalize different MSSOs and alleviate curse-of-dimension issues in SDDP, which has not been explored in BL-SDDP.
>
> Despite being orthogonal, we think that the BL-SDDP can be used in our framework to provide better supervision for our cut function prediction, and serve as an alternative for fine-tuning after $\nu$-SDDP-fast. We have included the above discussion in our refined manuscript.
>
> ### **"More thorough investigation… only two benchmarks are not sufficient to make such general statements"**
>
> We would like to first clarify that the two problems are from the real-world setting, where the structure of constraints, objective functions and dynamics are from the customer data. Due to the *privacy issue*, we use synthetically generated data in the inventory management case, while for portfolio we used the real-world US stock history price data in our model. Given that there’s no such benchmark for comparing ML based MSSO, we will release this real-world inspired benchmark dataset for future follow-up comparison. The new dataset is also one of our contributions to this under-explored research area.
>
> In our revision, we have made it clear that the conclusion is drawn based on the results from these benchmarks.
>
> ### **"What are the properties of the value function (in view of the piecewise elements) so that learning is effective?"**
>
> We have investigated following properties in the paper about the value function to reveal the cause of the effective learning:
> - Dimensionality: we’ve shown in [[Figure 4]](https://i.ibb.co/bNR7bWN/lowrank.jpg) that if the problem implicitly implies some low-rank structures (e.g., group of customer behaviors, seasonal similarities), then we can effectively learn the value functions (or cutting planes) in the low-dimensional space while almost maintaining the performance in the inventory management experiments.
> - Piecewise property: given that the value function is piecewise linear and convex, we have followed [Fullner & Rebennack](http://www.optimization-online.org/DB_FILE/2021/01/8217.pdf) (and also [De Matos et.al](https://www.sciencedirect.com/science/article/pii/S0377042715002794) as you mentioned) to use the lately added cutting planes as supervision, which shows better performance if the number of cuts is limited. We also show in [[Figure 3: Sml-Sht]](https://i.ibb.co/tz0F0pd/small-pieces.jpg) and [[Figure 3: Mid-Lng]](https://i.ibb.co/pzzgK04/large-pieces.jpg) that generally the hyper-network used in our work is able to capture different numbers of cutting planes, except when cutting planes in early iterations were used. This suggests that if the value function is provided in a high-quality and compact way, then it would benefit our training as well.
>
> ### **"Could … convergence of learning is so slow  … just adds an overhead?"**
>
> We would again clarify that we are working on an *meta-learning* setting, where we applied the learned neural network to new distributions of MSSO. Despite that training the neural network may take multiple hours due to the need of SDDP supervision, the inference of neural networks is much faster than doing one forward-pass (which involves LP solving) in SDDP, so the overhead regarding the runtime can almost be ignored.
>
> There could be situations when the test distribution of MSSO instances deviates a lot from the training ones, in which the learned value function predictor failed to generalize, then this may increase the number of iterations needed during fine-tuning with SDDP. We did not observe this in our empirical study, where the problems follow an underlying distribution. But we have made it clear in our revision in Appendix A about this potential limitation.

---

### Official Review · Reviewer_meaA · 2021-11-02

**Correctness:** 4
**Technical Novelty And Significance:** 2
**Empirical Novelty And Significance:** 3
**Recommendation:** 6
**Confidence:** 3

**Main Review:**

** Strengths:

- The paper presents a method that seems sound and non-trivial theoretically with good practical performance.


- Comparisons with RL algorithms are included.


- The paper is relatively easy to follow given its high degree of technical content

** Weaknesses:

- The subject area is of limited significance to ICLR community. MSSO does not have a history at top ML conferences and this paper does little to show it is a mistake.


- The experimental setting is very synthetic. The first problem is completely synthetic and as for portfolio optimization, I doubt there are real-world instances of portfolio optimization that do not optimize a quadratic objective that includes a risk term. This suggests that MSSO as a framework is potentially not rich and flexible enough to spark wider interest.


- Additionally, there are no strong baselines (e.g. coming from earlier benchmark-oriented research) available which makes it hard to judge how substantial the progress is.

** Suggestions

- What I believe would help this paper the most would be to invest in convincing the ML community of the relevance of the MSSO framework. If we can now indeed solve larger instances, can this be applied to any already established RL problems? Or is the linear objective too restrictive? Any steps in this direction would increase my assessment of the paper.

**Summary Of The Paper:**

The authors address the Multi-Stage Stochastic Optimization Problem. In this "MDP-like" setup, the underlying stochastic process generating state transitions is fixed (independent of actions) but the executed actions determine the subspace of action available in future steps. To solve this problem, the traditional approach is to solve a sequence of LPs, and "backpropagate" dual variables. This process includes heuristical choices of cutting planes. This paper improves uses a data-driven algorithm to improve this process. As a result, instances with a larger number of variables become manageable.



**Summary Of The Review:**

I believe the paper presents a sound and functional method that improves algorithmic performance on the MSSO problem. It is well-written and evaluated.

I remain unconvinced about the wider significance of MSSO but I am willing to listen to new arguments.

Overall, I have a mild tendency to primarily appreciate the technical quality and lean towards acceptance. However, at this point, I do not see myself championing the paper in case of a large variance of opinions.

---

> ### Author Response · Authors · 2021-11-23
> **Reply to Reviewer meaA's review**
>
> Dear reviewer, thanks for your valuable comments. Please see our response below:
>
> ### **ML relevance of MSSO**
>
> We have emphasized the importance of MSSO in the Introduction section and established the strong connection between MSSO and MDP/RL, which is a major community of ML in ICLR. We believe our work is providing RL community new modeling tools for sequential decision making with constraints (as in our side-by-side empirical comparison). Meanwhile our work introduces generally new practical applications of machine learning, specifically meta learning and representation learning, for accelerating and scaling up OR (stochastic optimization, combinatorial optimization, etc), which can be both important and interesting for machine learning and OR community.
>
> ### **The experimental setting**
>
> We would like to first clarify that the two problems are from the real-world setting, where the structure of constraints, objective functions and dynamics are from the customer data. Due to the *privacy issue*, we use synthetically generated data in the inventory management case, while for portfolio we used the real-world US stock history price data in our model.
>
> Given that there’s no such benchmark for comparing ML based MSSO, we will release this real-world inspired benchmark dataset for future follow-up comparison. The new dataset is also one of our contributions to this under-explored research area.
>
>
> ### **Portfolio optimization with risk term**
>
> We agree with the reviewer that our study has used a risk-neutral portfolio optimization as a case study and this simple risk profile plays a role in the optimal investment strategy. We would like to highlight that the goal of our experiment study is to show the solution quality and run time of our proposed $\nu$-SDDP and SDDP under a variety of problem setups and stochastic distributions, with risk-neutral portfolio optimization problem as one example provided in [1].
>
> However, we also like to note that SDDP has been extended to risk-averse setting as shown in [2], by considering the lower bound approximation of CVaR as the cost. Under this formulation, our method can be straightforwardly built upon the risk-averse SDDP for different risk profile problems in portfolio optimization, which is beyond the focus of this work.
>
> ### **Broad Interests of MSSO**
>
> We first emphasize that the major contribution of our paper is accelerating and scaling up the MSSO via a particular designed machine learning method, rather than proposing new planning models.
>
> We follow the standard formulation of MSSO in literature [3, 4, 5, 6]. We agree that the linear structure in both dynamics and objectives may be a limitation of applying in MSSO in practice. However, as we discussed in the maintext, MSSO has already been applied widely in inventory control, energy planning, and bio-chemical process control, where the dynamics are naturally linear. Moreover, we emphasize even with the linear structure, the computational complexity for solving MSSO is exponential w.r.t. either number of variables or number of horizons, which is highly non-trivial.
>
> ### **Strong baseline comparison**
>
> We first want to clarify that we made the comparison with the SDDP directly, which is the state-of-the-art solver. We show the time-solution trade-off in Figure 2 and more in Figure 7, 8 in appendix, where it shows we can quickly get reasonable solution quality while can still improve further.
>
> Moreover, we demonstrate the benefits of MSSO vs. RL methods by comparing the major existing deep RL methods, including DQN, DDPG, SAC, PPO. Those are strong baselines in the RL community, but they just didn’t perform well in this challenging task as these algorithms had a hard time dealing with the complicated hard constraints on action space and the model structures.
>
>
> >[1] George B Dantzig and Gerd Infanger. Multi-stage stochastic linear programs for portfolio optimization. Annals of Operations Research, 45(1):59–76, 1993\
> [2] Shapiro, Alexander, Wajdi Tekaya, Joari Paulo da Costa, and Murilo Pereira Soares. "Risk neutral and risk averse stochastic dual dynamic programming method." European journal of operational research 224, no. 2 (2013): 375-391.\
> [3] John R Birge. Decomposition and partitioning methods for multistage stochastic linear programs. Operations research, 33(5):989–1007, 1985.\
> [4] Mario V. F. Pereira and Leontina M. V. G. Pinto. Multi-stage stochastic optimization applied to energy planning. Mathematical Programming, 52(2):359–375, 1991.\
> [5] Alexander Shapiro, Darinka Dentcheva, and Andrzej Ruszczynski. ´ Lectures on stochastic programming: modeling and theory. SIAM, 2014.\
> [6] Guanghui Lan and Zhiqiang Zhou. Dynamic stochastic approximation for multi-stage stochastic optimization. Mathematical Programming, pp. 1–46, 2020.

---

> > ### Author Response · Authors · 2021-11-30
> > **Gentle reminder**
> >
> > Dear reviewer,
> >
> > Thanks again for your effort in reviewing the paper!
> >
> > We made the clarifications to try to resolve your concerns w.r.t the paper. This is a kind reminder to please check our response when you get a chance. We hope our clarification addressed your concerns, so that you may re-evaluate our submission. If you have more concerns about the paper, we would be more than happy to try to address them through this interactive rebuttal process.
> >
> > In any case, we highly appreciate your effort and your constructive suggestions that can potentially improve our paper quality!
> >
> > Best,
> >
> > Authors

---

### Official Review · Reviewer_hKKH · 2021-11-03

**Correctness:** 4
**Technical Novelty And Significance:** 3
**Empirical Novelty And Significance:** 3
**Recommendation:** 6
**Confidence:** 2

**Main Review:**

Strength:
- The paper is well-written, motivated, and organized. The comparison of MSSO and MDP makes the paper easier to follow for different communities. The motivation of the structure of the value functions is well explained.
- The proposed neural variant of SDDP addresses the exponential dependency of traditional SDDP over the size of the action space.
- The paper provides a thorough empirical investigation of their algorithm, which shows the advantage of $\nu$-SDDP.

Weakness:
- MSSO problem assumes that the problem context is completely known, while model-free RL does not require any knowledge on the model (such as reward function and transition kernel) and can directly learn from the data. It would be better if the paper could compare $\nu$-SDDP with RL methods with known MDP.
- Following the previous question, is it possible to solve MSSO via a completely data-driven approach? Specifically, the feasible action set is revealed at time t without knowing $A_t, B_t, b_t$.




**Summary Of The Paper:**

The paper studies the neural variant of stochastic dual dynamic programming for solving multistage stochastic optimization, which overcomes the limitation of SDDP. The paper proposes the algorithm $\nu$-SDDP, which continually self-improves by solving successive problems. The empirical investigation shows that $\nu$-SDDP outperforms SDDP and reinforcement learning in several synthetic and real-world process optimization problems.


**Summary Of The Review:**

The paper is well motivated. The proposed algorithm is novel and interesting. Thus, I recommend acceptance for the paper.

---

> ### Author Response · Authors · 2021-11-23
> **Reply to Reviewer hKKH's review**
>
> Dear reviewer, thanks for your valuable comments. Please see our response below:
>
> ### **$\nu$-SDDP v.s. RL methods with known MDP**
>
> Indeed, in our experiments, we compared the proposed $\nu$-SDDP with RL methods with known MDP. The deep RL algorithms are interacting with the **same** models as $\nu$-SDDP. We are considering the meta planning task in MSSO, where the model is given in each task as in the classic settings [1,2, 3, 4], while the model-free RL is naturally blind to the model information even the model information is given, and thus, is inferior as expected.
>
> ### **Solve MSSO via a completely data-driven approach**
>
> Thanks for the suggestion. Generally in real-world cases, the model is specified with clear customer needs (e.g., the inventory capacity, or the transportation cost) which are hard constraints that one should follow.
>
> In fact, equipped with the learned dynamics, the proposed method becomes a model-based RL method. However, as we discussed in the paper, even if the model is given, the MSSO is still difficult to solve. In this paper, we mainly focus on leveraging machine learning for the planner, while the model learning is out of the scope and can be an interesting future work.
>
>
> >[1] John R Birge. Decomposition and partitioning methods for multistage stochastic linear programs. Operations research, 33(5):989–1007, 1985.\
> [2] John R Birge and Francois Louveaux. Introduction to stochastic programming. Springer Science & Business Media, 2011.\
> [3] Mario V. F. Pereira and Leontina M. V. G. Pinto. Multi-stage stochastic optimization applied to energy planning. Mathematical Programming, 52(2):359–375, 1991.\
> [4] Alexander Shapiro, Darinka Dentcheva, and Andrzej Ruszczynski. ´ Lectures on stochastic programming: modeling and theory. SIAM, 2014.

---

> > ### Comment · Reviewer_hKKH · 2021-11-29
> > **Acknowledgement**
> >
> > Thank you for addressing my questions. I really appreciate it.

---

### Official Review · Reviewer_doAC · 2021-11-04

**Correctness:** 3
**Technical Novelty And Significance:** 3
**Empirical Novelty And Significance:** 3
**Recommendation:** 6
**Confidence:** 5

**Main Review:**

Strengths:

-The application to MSSO is interesting. There has been significant attention in using neural methods for deterministic optimization problems (integer programming, for instance). Given the utility of MSSO in OR problems, this learning-based framework could be impactful.

-The use of Earth mover’s distance in learning convex value functions is also interesting.

-There are some strong empirical results, showing significantly improved optimization times with little reduction in performance.

Weaknesses:

-Novelty of the overall framework is relatively limited. As I pointed out above, there have been many papers combining machine learning and existing optimization techniques and therefore I feel that the conceptual novelty of the current paper is low.

-How do RL algorithms, which focus on infinite horizon problems, get ported to the finite horizon case?

-The discussion of MSSO vs MDP is bizarre to me. It is accepted in the OR community that the MSSO formulation is essentially an MDP (see e.g. [1, 2]), where the state is (x_{t-1}, \epsilon_t) and the action is x_t.  In trying to understand the authors’ comment about using feasible sets as states, I’m wondering if perhaps they are working with a definition of “MDP” that doesn’t allow for state-dependent action sets? However, in many definitions of MDP (e.g., Sec 2.1.2 of Puterman’s text [3]), the set of feasible actions is explicitly allowed to depend on the state.

-In the conclusion of the paper claims that it works well in long-horizon settings, but In the inventory problem, problems of horizon 5 and 10 are considered (what is the horizon of the portfolio problem? I could not find it. Parameters for the portfolio problem should also be included in Table 1). In real-world instances, I would be worried that neural SDDP would need to accurately predict too many value functions to be practical. In such settings, perhaps infinite horizon MSSO models would be more appropriate approximation (see, e.g., [4, 5])? Discussion on this point and more evidence would improve the paper.

[1] ​​http://www.optimization-online.org/DB_FILE/2009/12/2509.pdf
[2] https://arxiv.org/abs/1605.01521
[3] https://onlinelibrary.wiley.com/doi/book/10.1002/9780470316887
[4] http://www.optimization-online.org/DB_FILE/2019/09/7367.pdf
[5] https://papers.ssrn.com/sol3/papers.cfm?abstract_id=2942921

**Summary Of The Paper:**

This paper considers a class of problems called multistage stochastic optimization (MSSO) (also known as stochastic programming) problems, which are essentially MDPs but with linear rewards and constraints. Such problems are very commonly studied in the OR community and solved using a technique called stochastic dual dynamic programming. The main contribution of the paper is a way to learn a neural mapping from MSSO problem instances to value functions, which can be used to warm-start the SDDP solver. Since the output of the neural network is a convex value function parameterized by a list of order invariant affine cuts, the authors propose using an Earth mover’s distance in their loss function.

**Summary Of The Review:**

My assessment is that the paper has a nice application to MSSO problems (which could be impactful due to their usage in OR models), but conceptually the novelty is low. The empirical work shows that the algorithms are generally impressive, but some details are missing. The paper could also use more clarity in writing -- see some of the comments that I brought up above.

---

> ### Author Response · Authors · 2021-11-23
> **Reply to Reviewer doAC's review**
>
> Dear reviewer, thanks for your valuable comments. Please see our response below:
>
> ### **Novelty**
>
> While there have been works leveraging machine learning to help optimization techniques in general, the integration with SDDP poses unique challenges where our new solution brings the technique innovation that enables:
> - the strict/exact treatment of constraints
> - alleviating the curse of dimensionality issue in SDDP, with the proposed contextual based optimal value function parameterized in piecewise linear function in principle component space
> - leveraging NN for predicting the convex function which brings the solution from Earth Mover’s distance, which avoids extra sampling in learning.
> - seamless integration with SDDP solver that allows further fine-tuning with vanilla SDDP algorithm.
>
> ### **How do RL algorithms used in infinite-horizon get ported to the finite horizon case**
>
> As analyzed in [1], in finite-horizon MDPs, we need to take the timesteps as input in state-action value function $Q(s, a; h)$ or policy $\pi(a|s; h)$. We included the parametrization of timesteps into state in the off-the-shelf RL algorithms to have non-stationary policies, as suggested in page 13 in [2]. We have added descriptions in the revised version.
>
> ### **The discussion of MSSO vs MDP**
>
> In fact, the introduced MDP reformulation of MSSO is essentially equivalent to the MDP form for MSSO built upon state-dependent action sets, as the reviewer pointed out. We on purpose rephrase it in this way to align with the current deep RL community. We acknowledge the reviewer’s suggestion and we have revised the equivalent form in the alternative way.
>
> ### **Long horizon case**
>
> The sample complexity of the SDDP algorithm grows linearly with the horizon length, but exponentially with the dimensionality. So we mainly focused on the dimension reduction technique in this case. Nevertheless, we include more results on the inventory management problem. We extend Mid-Lng setting in Table 1 with the same topology, but enlarge the horizon length $T$ to 50. We present the average error ratio below:
>
> | Horizon length | Average error ratio|
> |:-----------:|:--------:|
> | 10 |  3.29% |
> | 20 |  3.47% |
> | 30 |  3.53% |
> | 40 |  2.65% |
> | 50 |  0.82% |
>
> Overall in this setting it seems the $\nu$-SDDP-fast can still achieve a low error ratio compared to the optimal SDDP results. We have included these new results in Table 4 in appendix.
>
> ### **Portfolio optimization setting**
>
> We set the horizon to 5. We compared two settings: 5 stocks and 1000 stocks forming 8 clusters. More details can be found in Appendix E.2. We’ve updated Table 1 as you suggested.
>
> ### **Too-many value functions in reality and discussion on infinite-horizon MSSO**
>
> Our model relies on a hyper-network that is conditioned on the stage number and other context information for value function prediction, so computationally it would not cause much overhead regarding the runtime or memory in long-horizon situations.
>
> Thanks for the suggestion on infinite-horizon SDDP. Both infinite-horizon and truncated finite-horizon are the approximations of the real-world problems with extremely long planning stages.  In fact, our method can be straightforward extended to the setting with only **one** piecewise-linear function, which is relatively easier. In other words, the finite-horizon case might be more difficult than the infinite-horizon case. As our major contribution is introducing the learnable component into SDDP, we tested our algorithm in the difficult cases where we need to learn multiple optimal value functions for each stage.
>
> >[1] Puterman, Martin L. Markov decision processes: discrete stochastic dynamic programming. John Wiley & Sons, 2014.\
> [2] Schulman, John. "Optimizing expectations: From deep reinforcement learning to stochastic computation graphs." PhD diss., UC Berkeley, 2016.

---

> > ### Comment · Reviewer_doAC · 2021-11-29
> > **Response to authors**
> >
> > Thanks to the authors for the response. I've gone through it and it largely answers my questions and resolves many of the concerns. In particular, the explanations with horizon length help to convince me that the method can be practical. I'd like to increase my score.

---

### Decision · Program_Chairs · 2022-01-20

**Decision:**

Accept (Poster)

**Comment:**

This paper applies deep learning to a problem from OR, namely multistage stochastic optimization (MSSO). The main contribution is a method for learning a neural mapping from MSSO problem instances to value functions, which can be used to warm-start the SDDP solver, a state-of-the-art method for solving MSSO. The method is tested on two typical OR problems, inventory control and portfolio management. The reviewers think that the idea is interesting, the empirical results are impressive, and the paper is well-written. However, there are reservations on its relevance to the ICLR community.